# Deep Learning for Lung Cancer Diagnosis, Prognosis and Prediction Using Histological and Cytological Images: A Systematic Review

**DOI:** 10.3390/cancers15153981

**Published:** 2023-08-05

**Authors:** Athena Davri, Effrosyni Birbas, Theofilos Kanavos, Georgios Ntritsos, Nikolaos Giannakeas, Alexandros T. Tzallas, Anna Batistatou

**Affiliations:** 1Department of Pathology, Faculty of Medicine, School of Health Sciences, University of Ioannina, 45500 Ioannina, Greece; abatista@uoi.gr; 2Faculty of Medicine, School of Health Sciences, University of Ioannina, 45110 Ioannina, Greece; faybirbas@gmail.com (E.B.); kanavosus@gmail.com (T.K.); 3Department of Hygiene and Epidemiology, Faculty of Medicine, School of Health Sciences, University of Ioannina, 45110 Ioannina, Greece; gntritsos@uoi.gr; 4Department of Informatics and Telecommunications, University of Ioannina, 47100 Arta, Greece; tzallas@uoi.gr

**Keywords:** lung cancer, histopathology, histology, cytology, PD-L1, Digital Pathology, artificial intelligence, deep learning, convolutional neural networks, CNN

## Abstract

**Simple Summary:**

Lung cancer is one of the most common and deadly malignancies worldwide. Microscopic examination of histological and cytological lung specimens can be a challenging and time-consuming process. Most of the time, accurate diagnosis and classification require histochemical and specific immunohistochemical staining. Currently, Artificial Intelligence-based methods show remarkable advances and potential in Pathology and can guide lung cancer diagnosis, subtyping, prognosis prediction, mutational status characterization, and PD-L1 expression estimation, performing with high accuracy rates. This systematic review aims to provide an overview of the current advances in Deep Learning-based methods on lung cancer by using histological and cytological images.

**Abstract:**

Lung cancer is one of the deadliest cancers worldwide, with a high incidence rate, especially in tobacco smokers. Lung cancer accurate diagnosis is based on distinct histological patterns combined with molecular data for personalized treatment. Precise lung cancer classification from a single H&E slide can be challenging for a pathologist, requiring most of the time additional histochemical and special immunohistochemical stains for the final pathology report. According to WHO, small biopsy and cytology specimens are the available materials for about 70% of lung cancer patients with advanced-stage unresectable disease. Thus, the limited available diagnostic material necessitates its optimal management and processing for the completion of diagnosis and predictive testing according to the published guidelines. During the new era of Digital Pathology, Deep Learning offers the potential for lung cancer interpretation to assist pathologists’ routine practice. Herein, we systematically review the current Artificial Intelligence-based approaches using histological and cytological images of lung cancer. Most of the published literature centered on the distinction between lung adenocarcinoma, lung squamous cell carcinoma, and small cell lung carcinoma, reflecting the realistic pathologist’s routine. Furthermore, several studies developed algorithms for lung adenocarcinoma predominant architectural pattern determination, prognosis prediction, mutational status characterization, and PD-L1 expression status estimation.

## 1. Introduction

Lung cancer is one of the most prevalent cancers worldwide, characterized by a high mortality rate, reaching up to 18% of total cancer-related deaths, with cigarette smoking being the leading cause [1]. Lung cancer is a heterogeneous disease, mainly classified as non-small cell lung carcinoma (NSCLC) and small cell lung carcinoma (SCLC) [2]. NSCLC constitutes the majority of lung cancer cases (85%) and is further classified into adenocarcinoma (ADC), squamous cell carcinoma (SCC), and large cell carcinoma (LCC), while the remaining 15% accounts for SCLC, which is characterized by neuroendocrine differentiation.

In the era of personalized medicine, lung cancer diagnosis and accurate classification strongly rely on cytological and histological subtyping by microscopic evaluation with standard histochemical stains and ancillary immunohistochemical staining [3]. Molecular testing is also necessary for personalized therapeutic targeting and monitoring for patients’ stratification to targeted therapy and immunotherapy [4,5]. According to published guidelines by the College of American Pathologists, the International Association for the Study of Lung Cancer, and the Association for Molecular Pathology, patients with advanced lung cancer with an ADC component should be tested for epidermal growth factor receptor (EGFR) mutations, anaplastic lymphoma kinase (ALK) and ROSproto oncogene 1 (ROS-1) rearrangements, v-Raf murine sarcoma viral oncogene homolog B (BRAF) Val600Glu (BRAFV600E), Ret Proto-Oncogene (RET) rearrangements, mesenchymal-epithelial transition (MET) exon 14 skipping mutations, Kirsten rat sarcoma (KRAS) mutations, and neurotrophic tyrosine kinase receptor fusions (NTRK1-3) [2,6]. Advanced-stage non-neuroendocrine carcinomas should be tested for programmed cell death ligand 1 (PD-L1) expression status as patients with a PD-L1 Tumor Proportion Score (TPS) ≥ 50% are eligible for first-line treatment with the anti-PD-L1 therapy, pembrolizumab. Immunohistochemical assays are available for PD-L1 and ALK expression status detection [7,8,9,10]. Currently, reflex-ordered testing for lung cancer is gaining ground, underlining the necessity of collaboration between pathologists and oncologists. Although reflex testing is not feasible to perform in many laboratories, it can provide additional valuable information, detect rare molecular alterations, and minimize testing turnaround time [3,11].

In the last decade, Deep learning (DL) approaches, including mostly Convolutional Neural Networks (CNNs), are increasingly valuable in Pathology. Limitations concerning the shortage of pathologists worldwide, subjectivity in diagnosis, and intra- and inter-observer variability could be overcome with the aid of DL models. Recent advances in lung cancer pathology leverage image analysis potential for cancer diagnosis from hematoxylin and eosin (H&E) whole slide images (WSIs) [12,13]. Considering that small biopsy and cytology specimens are the available material for 70% of lung cancer patients with advanced unresectable disease, DL methods could guide the diagnosis with high accuracy, minimizing the need for additional special stains required for differential diagnosis and preserving the already limited material for molecular testing [2,14,15].

In this review, we systematically outline the current implications of DL algorithms for lung cancer diagnosis, prognosis, and prediction using both histological and cytological images. We further summarized the extracted data into distinct categories based on the classification problem, presenting for each study the dataset details, the employed technical method and methodology, as well as the performance metrics. The different categories have been structured to be informative for both pathologists and cytologists, can provide a detailed analysis and a comprehensive guide of the existing DL applications for lung cancer, and offer valuable information to researchers for further study.

## 2. Materials and Methods

The systematic review followed the recommendations of the Preferred Reporting Items for Systematic Reviews and Meta-Analyses (PRISMA) [16]. The protocol has not been registered.

### 2.1. Search Strategy

We systematically searched PubMed from inception to 31 March 2023 for primary studies developing a DL model for the histopathological or cytopathological interpretation of malignant lesions in lung specimens. For this purpose, we used the following algorithm: (convolutional neural network* OR CNN OR deep learning) AND lung AND (cancer OR neoplas* OR carcinoma OR adenocarcinoma OR malign* OR tumor*) AND (histolog* OR histopatholog* OR eosin OR slide* OR section* OR immunohistochem* OR biop* OR microscop* OR cytolog* OR cytopatholog* OR immunocytochem*).

### 2.2. Study Eligibility Criteria

Eligible articles were considered on the basis of the following criteria. We included studies, the aim of which was to develop at least one DL model for the histopathological or cytopathological assessment of malignant lesions in lung specimens. Eligible applications of the DL models included diagnosis, subtyping, prognosis evaluation, characterization of mutational status, and prediction of PD-L1 expression. Articles that presented in vitro models or used non-histological/cytological data as well as reviews/meta-analyses, editorials, letters, and invited opinions, were excluded. In addition, articles not available in English and those referring to organisms other than humans were deemed ineligible.

### 2.3. Study Selection

All citations collected by the previously mentioned methodology were independently screened by four authors, who were properly trained before the process started, using the Rayyan web application [17]. Three of these researchers were scientifically capable of evaluating the medical aspect of the query, and one of them was a CNN expert, able to assess the technical part. During the screening period, the researchers met regularly to discuss disagreements and continue training. Conflicts were resolved by consensus. The full texts of potentially eligible articles were later retrieved for further evaluation.

### 2.4. Data Extraction

To facilitate the data extraction process, we specially designed a spreadsheet form, which all researchers could access to import data from all the eligible articles. From each paper, we manually extracted information regarding the first author, year of publication, aim of medical research, technical method, classification details, dataset, and performance metrics.

## 3. Results

Our systematic search returned 357 articles, 127 of which were selected for full-text assessment. Ultimately, 96 articles met our criteria of eligibility and were included in our study. A detailed description of the study selection process can be found in the PRISMA flow diagram presented in Figure 1.

At first, the included studies were divided, based on the used dataset, into histology and cytology sections. Further categorization of the histology section into diagnosis, lung cancer classification, NSCLC subtyping, ADC predominant architectural patterns classification, prediction of prognosis and survival, and prediction of molecular alterations subsections was made based on the classification problem. Studies performing DL for the PD-L1 expression status estimation were summarized in a particular section.

### 3.1. Histology

#### 3.1.1. Diagnosis

Jain et al. used DL architectures for detecting lung cancer from histopathological images pre-processed for size, normalization, and noise removal [18]. Three datasets were included achieving high-performance rates with an accuracy of over 97%. Jiao et al. proposed a rapid and efficient method for tumor classification called Deep Embedding-based Logistic Regression (DELR) [19]. DELR was applied in three different datasets (colorectal, lung, and breast cancer) and achieved an area under the curve (AUC) of over 0.95 for all three datasets. In lung cancer, the dataset consisted of 338 regions of interest (ROIs), including ADC and SCC images. Moreover, Kanavati et al. trained a CNN to distinguish lung carcinoma from non-neoplastic tissue based on the EfficientNet-B3 architecture [20]. After training, the CNN was tested on four independent datasets and attained an AUC of more than 0.97, demonstrating its feasibility of generalization. Multiple Instance Learning (MIL) was employed for the same classification task without the need for manual annotations by pathologists [21]. A multi-organ classification using weakly supervised learning was performed by Tsuneki et al. on transbronchial lung biopsy WSIs [22]. The AUC values of the three different balanced training datasets collected from medical institutions were 0.879–0.933 (Table 1)**.**

#### 3.1.2. Lung Cancer Classification

A common classification problem among all papers included refers to lung cancer tissue classification into the main categories of ADC, SCC, and SCLC according to WHO guidelines. Kanavati et al. developed a CNN for lung cancer subtyping (ADC, SCC, SCLC, and non-neoplastic tissue) trained on transbronchial biopsy (TBLB) images with mainly poorly differentiated carcinomas [24]. Their model was tested on four validation cohorts (one with TBLB specimens and three with surgical resections), performing with AUC over 0.9 on all datasets. The same classification problem was employed with weakly supervised CNN, including a smaller dataset from hospital archives and The Cancer Genome Atlas (TCGA) database [25]. The model had an overall accuracy of 97.3% and achieved an AUC of 0.856 in the TCGA cohort. In addition, three common CNNs (Inceptionv3, VGG-16, InceptionResNetV2) were used for lung cancer classification on TMAs. The InceptionV3 model achieved the highest performance; however, many cases of ADC and SCC were misclassified [26]. In a retrospective study by Yang et al., a six-type classifier model was designed for lung cancer (ADC, SCC, SCLC) as well as other lung diseases (pulmonary tuberculosis, organizing pneumonia) subtyping on H&E-stained slides [27]. The proposed classification task achieved great performance and consistency with experienced pathologists. In a different study, Yang et al. introduced a CNN for subtyping lung cancer in five classes, namely ADC, SCC, SCLC, large cell neuroendocrine carcinoma (LCNEC), and non-tumor [28]. The customized model performed similarly or better than the pre-trained ones, although existing limitations of the study, such as the use of patches instead of WSIs and the limited dataset, resulted in moderate classification accuracies. Likewise, Kosaraju et al. applied a novel DL framework for classifying ADC, SCC, SCLC, and LCNEC, achieving an AUC of 0.96 [29]. The studies of Yang and Kosaraju et al. were the only ones that included LCNEC in the classifiers representing the realistic diagnostic practice for a pathologist. Ilié et al. applied a DL algorithm for distinguishing SCLC, LCNEC, and atypical carcinoid (AC) [30]. A number of 150 H&E WSIs were included, and the model was in great agreement when compared to expert and general pathologists, achieving an AUC of 0.93. Lastly, in their recent study, Chen et al. proposed an immunohistochemical phenotype prediction system for upgrading the classification of lung cancer into ADC, SCC, and SCLC [31]. The WSI-based Immunohistochemical Feature Prediction System (WIFPS) discriminated lung cancer types on H&E slides based on the positive or negative expression scoring of characteristic biomarkers for each class (ADC: TTF-1, CK7, and Napsin-A; SCC: CK5/6, p40, and p63; SCLC: CD56, Synaptophysin, Chromogranin A, and TTF-1). The agreement between the WIFPS model and pathologists achieved high to almost perfect consistency (Cohen’s kappa value of 0.7903–0.8891) in validation sets and the AUC in surgical and biopsy images was over 0.8 in all validation cohorts. In addition, ALK prediction status achieved an AUC of 0.917; however, programmed cell death protein 1 (PD-1), PD-L1, KRAS, and EGFR status did not reach high performance (Table 2).

#### 3.1.3. NSCLC Subtypes Classification

The diagnosis between ADC and SCC from a single H&E slide from a small biopsy or cytological material can be challenging. Thus, for precise diagnosis, additional staining for immunohistochemical biomarkers, such as TTF-1, CK5/6, CK7, pan keratin, p40, and p63, and histochemical stains, such as periodic acid-Schiff (PAS), must be performed. Several studies have addressed binary classification problems concerning NSCLC subtyping from H&E slides for an accurate and fast diagnosis. The majority of these mainly include ADC and SCC WSIs, mostly from the TCGA dataset, whereas the classification task is performed by a CNN or a combination of the state-of-the-art CNN architectures with varying approaches and techniques [37,47,51,52,53,54,55,56,58,59]. Moreover, NSCLC subtyping was combined with genomic data, namely copy number variations (CNVs), from TCGA [42]. The authors demonstrated that their proposed LungDIG model could be of great importance not only for ADC and SCC diagnosis but also for stratifying patients for targeted therapies, as the performance metrics of the model were higher when WSI and CNV data were combined compared to when WSI or CNV features were used alone. Zhao et al. developed a weakly supervised DL model to localize ROIs on WSIs (AUC of 0.9602) and then accurately subtype NSCLC into ADC and SCC with high sensitivity and specificity rates (0.9474 and 0.8583, respectively) [43]. In another study extracting prominent deep features (DFs) for each histopathological image, classification accuracy was better, and the authors identified 15 DFs with the ability to classify lung cancer with an accuracy of over 85% [44]. The generalizability of the model was feasible in distinguishing ADC from SCC on 21 non-pulmonary carcinomas; however, classification accuracy reached 56% in the external validation cohort. Hou et al. performed a classification task of NSCLC subtyping into ADC, SCC, and ADC with mixed subtypes [57]. Their proposed framework was trained and tested on a TCGA dataset with a classification accuracy of 0.798. Masud et al. designed a classification framework for diagnosing lung and colon cancer from histopathological images from the LC25000 dataset [49]. The model achieved a peak classification accuracy of 96.33%; however, the lung ADC class had a higher misclassification rate. The same problem using the LC25000 dataset was employed by other authors, with an overall accuracy of 99% [32,33,36,39]. DarkNet-19 model reached accuracies of 97.57%, 99.87%, and 97.73% in classifying ADC, benign, and SCC images, respectively, while the overall accuracy of the model was 99.69% [45]. Likewise, Civit-Masot et al. employed Explainable Artificial Intelligent (AI) Technologies [23]. Liu et al. used AI along with activation function for cancer infiltration screening on histopathological images [38]. Their method was further utilized for lung cancer classification (ADC and SCC) using the LC25000 database, presenting good generalization ability. In a more recent study, Liu et al. proposed a novel method for automated detection of lung ADC infiltration using 780 images with sensitivity and specificity of 93.10% and 96.43%, respectively [60]. Utilizing a combination of molecular and histological data (gene expression data and WSIs, respectively) as input for NSCLC classification, Carrillo Perez et al. demonstrated that the fusion model could provide robust information for decision-making to targeted therapies [46]. Wang et al. proposed a platform for the automated classification of NSCLC into ADC, SCC, and normal regions as well as for prediction of mutational status of 10 frequently mutated genes in ADC [50]. The model predicted with an AUC of 0.824 the EGFR mutational status on ADC H&E WSIs. Similarly, a model for NSCLC subtyping (ADC, SCC, normal regions) achieved an AUC of 0.97 [56]. The authors trained the model to predict the mutational status in lung ADC slides. Of the ten frequently mutated genes in ADC, STK11 and KRAS had the highest AUC (0.845 and 0.814, respectively). An annotation-free DL method for the subtyping of NSCLC slides achieved high performance for ADC and SCC (AUC of 0.9594 and 0.9414, respectively) and could be employed in clinical practice as it overcomes the time-consuming process of annotations and limitations concerning the capacity/memory of WSIs [48]. Wang et al. developed a DL model to perform cancer lesion region segmentation and histological subtype classification on ADC and SCC slides [40]. The model showed high classification performance metrics (accuracy was 100% and 95.1%, sensitivity was 95.0 and 100.0%, and specificity was 95.2 and 100.0% for SCC and ADC classification tasks, respectively). Classification of transcriptomic lung ADC (bronchioid, squamoid, and magnoid) and/or SCC (primitive, classic, secretory, and basal) subtypes was performed by Yu and Antonio et al. [53,61]. In the first study, classification was performed on both ADC and SCC, resulting in a significant correlation between the transcriptomic subtype and the histopathology classification scores and achieving AUC of 0.771–0.892 and approximately 0.7 for ADC and SCC, respectively, with the employment of four CNNs. In the study of Antonio et al., ADC transcriptome subtype classification resulted in a classification accuracy of 98.9%. Lastly, Le Page et al. tried to distinguish squamous from non-squamous lung carcinoma from initial cytology and small biopsy specimens [41]. Their model performed with good classification accuracy, while the accuracy was slightly increased in the external validation cohorts when tissue microarrays (TMAs) were selected (accuracy rates of 0.78 in biopsies versus 0.82 in TMAs). Finally, two recent studies performed a binary classification between ADC and SCC using over 900 WSIs from TCGA and achieving an AUC of over 0.90 [34,35] (Table 2).

#### 3.1.4. Lung ADC Predominant Architectural Patterns Classification

ADC cases exhibit various histological patterns. According to the WHO, there are five distinct histological subtypes (lepidic, acinar, papillary, micropapillary, and solid) that must be included in a pathology report when the material is a resection specimen [2]. The detection of ADC predominant architectural patterns has been the scope of several research papers (Table 3)**.** The study by Sadhwani et al. performed a classification problem including six histological subtypes (acinar, lepidic, solid, papillary, micropapillary, cribriform) and then combined the predicted output with clinical data (smoking status, age, etc.) for tumor mutational burden (TMB) status prediction [62]. The AUC for ADC predominant architectural patterns classification was 0.93 and 0.92 for TCGA and the external validation cohort, respectively, while for the TMB status prediction, it was 0.71. Furthermore, a six-class problem (lepidic, acinar, papillary, micropapillary, solid, benign) for lung ADC histological subtypes classification in lung ADC WSIs was in moderate agreement with pathologists’ estimations [63]. In a similar study, ADC histological patterns were classified into five categories (solid, micropapillary, acinar, cribriform, non-tumor) using three different CNN architectures [64]. The best classification accuracy was 89.24%, while, in the study of DiPalma et al., the histological classification of the known five patterns of lung ADC resulted in a classification accuracy of 94.51% [65]. Xiao et al. created a novel framework combining CNNs and graph convolutional networks for quantitative estimation of histopathological growth patterns in lung ADC slides [66]. Another lung ADC subtyping problem was performed by Sheikh et al. achieving a high accuracy rate of 0.946 and outperforming the state-of-the-art models [67]. In a different study, Gao et al. collected slides from ADC with micropapillary patterns and performed a binary classification problem for detecting the presence of a micropapillary pattern in ADC slides [68]. Maleki et al. investigated how several possible methodological errors, such as oversampling and data augmentation, can lead to poor generalizability performance and performed a binary classification task for the distinction of solid and acinar predominant histologic subtypes in ADC H&E slides [69].

#### 3.1.5. Prediction of Prognosis and Survival

The quantification and evaluation of the tumor microenvironment (TME) features from histopathological images, derived by the spatial distribution of different cell types (lymphocytes, stromal cells), the density of stromal cells, etc., provide valuable information not only for immune therapy response but also for the probability of survival [70]. TME plays an important role in immunotherapy response as well as in cancer progression and metastasis in lung cancer. Several studies have aimed to develop algorithms for TME characterization of lung cancer pathology images to predict response to targeted therapies and extract prognostic value. Barmpoutis et al. proposed a methodology to identify and quantify tertiary lymphoid structures (TLS) in lung cancer H&E images [71]. Segmentation of lymphocytes showed that their density within a TLS region was 3-fold higher than lymphocytes outside TLS regions. Their study had high sensitivity and specificity rates and could be used as a prognostic feature to predict response to immunotherapy. DeepRePath was proposed for prognosis prediction in patients with early-stage ADC [72]. On the external validation cohort, DeepRePath had an AUC of 0.76, while histopathological features, such as necrosis or atypical nuclei, were associated with a higher probability of recurrence. The same model, DeepRePath, was employed by Wu et al. for predicting the recurrence risk of lung cancer, achieving an AUC of 0.79 on a small testing cohort [73]. In the study of Wang et al., cell type classification into tumor cells, stromal cells, and lymphocytes achieved great classification accuracy [74]. TME analysis for spatial distribution estimation associated TME with overall survival (OS) and could provide valuable information about the patient’s prognosis. In a similar framework, Wang et al. proposed a CNN for a 6-class classification problem to identify different cell types nuclei for estimating TME and its prognostic value [75]. The derived features from the TME analysis were indicators of OS. For instance, higher karyorrhexis density was associated with worse survival outcomes, while higher stromal nuclei density was associated with better survival outcomes. Moreover, segmentation of cell nuclei on H&E WSIs was applied to identify and quantify tumor-infiltrating lymphocytes (TILs) for prognostic value on NSCLC patients [76]. The authors highlighted the potential of their proposed model for quantifying TILs, instead of immunohistochemical staining (CD8), for assisting pathologists. Likewise, the quantitative and spatial localization characteristics of TILs and tumor cells were evaluated for OS and relapse-free survival (RFS) in NSCLC cohorts [77]. From 10 immune checkpoint proteins, galectin-9 and OX40L had the higher relative contribution to OS (33.55%) and RFS (29.02%), respectively, while the percentage of positive tumor cells and the distance between positive TILs and positive tumor cells contributed the most to predict OS. A two-step approach of a DL method was proposed by Pham et al. for detecting lung cancer lymph node metastasis [78]. The proposed approach was developed to eliminate false positive results by performing a first classification task for distinguishing reactive lymphoid follicles from lung cancer in lymph nodes. In the study of Rączkowski et al., tumor prevalence and TME composition were used as input for predicting survival and gene mutations in lung ADC cases [79]. The prediction of OS on the lung dataset was evaluated according to clinical and demographic data [80]. The proposed weakly supervised and annotation-free CNN achieved a C-index of 0.7033, and features such as TILs, necrosis, and inflamed stromal regions were identified as prognostic factors associated with poor outcomes. Estimation of lung ADC tumor cellularity for genetic tests by pathologists could be improved by DL support. Sakamoto et al. showed that tumor cellularity can be estimated with minimum deviation from the ground truth when pathologists and AI scores are combined [81]. Pathologists’ estimations deviated from the ground truth by approximately 15%, implying over- or under-estimations; however, false positive results were obtained by AI when cell blocks were evaluated. Prediction of lung ADC recurrence in several predominant subtypes, including acinar and papillary carcinoma, after complete resection achieved an accuracy of 90.9% in H&E WSIs from 55 patients [82]. The density of cancer epithelium and cancer stroma lymphocytes was calculated in H&E slides from lung ADC cases to predict patients’ survival [83]. Low score rates were associated with significantly superior OS and disease-free survival in patients with ADC. The authors also included RNA transcripts to determine the TILs infiltration between the high-risk and low-risk groups revealing that patients in the low-risk group had a higher proportion of CD8+ T cells, activated CD4+ memory T cells, and plasma cells versus those in the high-risk group. Slides of lung ADC immunohistochemically stained for CD3, CD8, and CD20 were used for the detection and quantification of immune cell biomarkers [84]. High sensitivity and specificity rates were recorded in discriminating T cells, considering the immunostaining intensity variables and the presence of anthracotic pigment in the tissue slides. In a recent study, a DL method was employed for predicting aneuploidy from lung ADC WSIs performing nuclei segmentation and using a single-cell analysis [85] (Table 4).

#### 3.1.6. Prediction of Significant Molecular Alterations

Molecular detection of prognostic and predictive biomarkers in specific histological subtypes can predict favorable responses to targeted therapy and treatment. The detection of significant molecular alterations on immunohistochemistry (IHC) slides using DL algorithms was the scope of several studies. Concerning ALK rearrangements prediction, in the study by Terada et al., the commercially available HALO-AI platform and DenseNet were employed in IHC slides achieving a maximum AUC of 0.73 (in the resolution of 1.0 µm/pix) [89]. Another study aimed to predict mutations (EGFR, BRAF, TP53, STK11, and KRAS) based on Next Generation Sequencing (NGS) data and H&E WSIs from ADC samples with several deep neural network-based models [90]. Predicting EGFR and TP53 mutations achieved better performance compared to the remaining genes involved in the study. In the study of Wang et al., the proposed model for predicting the mutational status of 10 frequently mutated genes in ADC slides had the best performance for EGFR mutational status with an AUC of 0.824 [50]. Similarly, Coudray et al. trained the Inceptionv3 network to predict the mutational status of 10 genes in lung ADC, with STK11 and KRAS having the highest AUC of 0.845 and 0.814, respectively [56]. In addition, high performance was recorded for TP53 and EFGR biomarkers prediction, with AUC of 0.87 and 0.84, respectively, in the study of Yang et al. [87]. However, the model was not validated on an external cohort, and only 180 WSIs from the TCGA database were used. MET, FGFR1, and FGFR2 mutations were predicted with accuracies of 86.3%, 83.2%, and 82.1%, respectively [91]. The recent study by Mayer et al. was the first to employ DL for predicting ROS1 rearrangement directly from H&E WSIs [92]. ROS1 rearrangement prediction reached sensitivity and specificity of 100% and 98.48%. Moreover, the characterization of intra-tumor heterogeneity in ADC by gene expression levels was associated with patients’ survival [86]. In the lung cancer dataset, the highest AUC was detected for miR-17-5p microRNA, followed by KRAS and CD274 (PD-L1). Another study determined TMB value (low or high) according to a selected threshold in lung ADC WSIs. TMB value was predicted for each area of the image, reflecting the heterogeneity of TMB [93]. No significant correlation between the TMB status and the tumor stage of the patient was noted, while the performance of the DL model was relatively low, with an AUC of 0.641. Likewise, the prediction of TMB in 50 SCC H&E images achieved an AUC of 0.65 (Table 5) [94].

### 3.2. Cytology

Cytological specimens from the lung are frequently the only available diagnostic material. However, by its nature, this material is limited, prohibiting auxiliary techniques for specific subtyping, such as immunocytochemistry. Only a limited number of studies have addressed the issue of utilizing cytological images for training neural networks for lung cancer diagnosis and subtyping (Table 6). The first study for the classification of lung cancer cytological images (ADC, SCC, SCLC) achieved a classification accuracy of 71% after the data augmentation process [97]. In addition to this study, Teramoto et al. further extended their work for the classification of lung cytological images (real and synthesized) into benign and malignant with a generative adversarial network (GAN) [98]. The proposed method achieved an AUC of 0.901. Similarly, the classification of benign and malignant cells from cytological pleural effusions WSIs, by a weakly supervised model achieved an AUC of 0.9526 [99]. The model had a significantly strong correlation with the histological diagnosis gold standard as well as with senior cytopathologists’ diagnosis. Misclassification was observed when poor adhesion of tumor cells or clusters of mesothelial cells were present. Diagnosis between benign and malignant cells from cytological specimens was performed in the studies of Lin and Teramoto et al., including 499 and 322 images, respectively [100,101]. Distinct morphological features (size of cells, nuclei, and nucleoli) of cytological specimens of lung cancer were recognizable by four different fine-tuned deep CNNs (DCNNs) [102]. Three out of four DL models resulted in a classification accuracy of more than 73% for lung cancer subtyping into ADC, SCC, and SCLC; however, some cases of poorly differentiated NSCLC were misclassified. Furthermore, the distinction between LCNEC and SCLC showed promising results in the study of Gonzalez et al. [103]. Three classifiers were developed with three distinct datasets of Diff-Quik^®^-, Papanicolaou- and H&E-stained cytological WSIs and achieved an AUC of 1, 1 and 0.875, respectively. Lastly, endobronchial ultrasound (EBUS)-guided transbronchial needle aspiration (TBNA) cytological images were employed for diagnosing mediastinal metastatic lesions [104]. The study by Wang et al. was the first to include EBUS-TBNA cytological images for automatic segmentation of enlarged mediastinal lymph nodes metastasis, outperforming three state-of-the-art baseline models.

### 3.3. PD-L1 Expression Status

PD-L1 is an immune checkpoint protein expressed on tumor cells and activated immune cells [105]. In NSCLC patients, assessment of PD-L1 expression is pivotal for guiding patients’ treatment selection with immune checkpoint inhibitors (ICIs). IHC is the currently accepted diagnostic assay performed on formalin-fixed paraffin-embedded (FFPE) lung tissue or cytological specimens [106]. There are different platforms for IHC interpretation, PD-L1 antibodies, guidelines for evaluation and scoring, as well as positivity cut-offs for immunotherapy selection. Currently, four IHC assays (28-8 and 22C3 from DAKO, SP263 and SP142 from Ventana) have been approved for use by the Food and Drug Administration (FDA). The 22C3 and 28-8 pharmDx (DAKO) IHC assays are companion diagnostics for selecting patients for pembrolizumab and nivolumab, respectively [107,108]. SP142 and SP263 (Ventana) IHC assays are also FDA-approved for companion diagnostic to atezolizumab. Evaluation of PD-L1 expression with the 22C3 and 28-8 pharmDx, as well as SP263 (Ventana) assays, only refers to the PD-L1 expression on tumor cells, while, on the other hand, the SP142 (Ventana) assay refers to tumor and immune cells staining [109]. As PD-L1 scoring algorithms determine the therapeutic choice and interobserver discordance is common, it is conceivable that quantitative validation of PD-L1 expression by DL algorithms may assist pathologists in their assessment. In a recent study, Hondelink et al. developed a fully supervised DL model for PD-L1 TPS assessment in NSCLC WSIs according to three cut-off points (<1%, 1–50%, and 50–100%) [110]. TPS prediction was in concordance with the mean score of three pathologists in 79% of the cases. Misclassification of some cases was noted when positive PD-L1 immune cells were present around the tumor site, the intensity of PD-L1 positive neoplastic cells was weak, or when non-membranous staining was detected. In a similar framework, Liu et al. performed tumor region segmentation and nuclei detection for PD-L1 TPS prediction on SCC WSIs according to three cut-off points (<1%, 1–49%, and ≥50%) [111]. Their proposed model’s predictions were compared to the pathologist’s prediction with different experience levels. The model’s classification accuracy was 74.51%, higher than trainees (71.55%) but lower than subspecialist and non-subspecialist pathologists (97.06% and 84.03%, respectively). In another study, TPS assessment reached high performance in terms of sensitivity and specificity in both 1% and 50% cut-off points [112]. The classification was performed on slides stained with 22C3 antibody, and the proposed patch-based dual-scale categorization method based on VGG16 architecture achieved higher performance compared to VGG16. The study of Sha et al. resulted in an AUC of 0.80 on a balanced testing cohort in classifying positive and negative PD-L1 tumor cells [113]. In SCC-separated cases, the model achieved a lower AUC compared to ADC cases (0.64 and 0.83, respectively), maybe due to an imbalance in the training cohort. In the studies of Kapil et al., TPS was estimated by dividing the pixel number of positive tumor cells by the total pixel number of positive and negative tumor cells [114,115]. Of all the included studies estimating PD-L1 TPS, these two were the only ones using slides stained with SP263 antibody with the cut-off point defined at 25%. In their first study, fully- and semi-supervised network architectures were used for estimating TPS in NSCLC specimens, with results agreeing with pathologists’ evaluation, while, in their subsequent study, TPS estimation was performed with a GAN. Two classification problems were addressed, namely, a binary task for epithelial and non-epithelial region segmentation as well as TPS estimation. An additional dataset of WSIs stained with the epithelial marker Pan-Cytokeratin was used for the binary segmentation task of the epithelial benign and malignant regions. In the study of Wu et al., PD-L1 IHC slides stained with 22C3 assay were used for training U-Net to perform tumor area detection and TPS calculation [116]. The model was highly consistent with trained pathologists and achieved high performance when further tested in SP263 (Ventana) stained slides (accuracy of 0.9326 and 0.9624 for 22C3- and SP263-stained slides, respectively). Furthermore, the authors demonstrated that the AI-based model could help untrained pathologists with TPS assessment by reducing the time of microscopic examination. In the same framework, three automated workflows based on DL, including both 22C3 (DAKO) and SP263 (Ventana) IHC assays, and two cut-off points (<1%, ≥50%), achieved better performance in the <1% cut-off point [117]. The model by Choi et al. achieved an area under the receiver operating characteristic (AUROC) of 0.889 in detecting PD-L1 positive and negative tumor cells and estimating TPS value, while it significantly increased the concordance of pathologists after a disagreement (initial/baseline concordance of 81.4% versus revised concordance of 90.2%) [118]. Aitrox’s AI performance for PD-L1 expression by Huang et al. was comparable to those of experienced pathologists, while it surpassed inexperienced ones (Table 7) [119].

### 3.4. Deep Learning Approaches

From a clinical point of view, the main challenges in Digital Pathology are (i) the extremely large size of the images produced by whole-slide scanning and the requirement for pathologists to evaluate the entire specimen; (ii) the digitization of annotated findings of interest, which is a very demanding and time-consuming process. The latter, combined with the fact that DL techniques require a large amount of training data, intensifies the problem of the provision of reliable results. Many studies presented in the literature attempt to overcome the lack of annotations by using weakly supervised or semi-supervised learning techniques instead of fully supervised approaches. These approaches interact with known CNN architectures to classify patches of images or to detect tissue alterations and/or morphological features of cancer. Weakly supervised learning is a branch of machine learning (ML) that aims to use less or lower quality labels for training predictive models. It works by leveraging the unlabeled data or refining the labels to improve the model performance. In terms of Digital Pathology, weakly supervised methods use a small number of annotations by selecting informative patches to classify the WSIs [25]. General approaches of weakly supervised learning in histopathological images have been proposed, employing VGG-16 [25], EM-CNN [52], EfficientNet-B3 [20], and ResNet [80]. Furthermore, most of the presented studies in this category employ MIL [34,85], which is a weakly supervised learning technique that groups data points into bags. Each bag is labeled with the class by the instance count of that particular class. This technique is well-suited for histology slide classification because it is designed to operate on weakly-labeled images [65]. For example, clustering-constrained-attention MIL (CLAM), developed by Lu et al. [47], is a weakly supervised method that uses attention-based learning to automatically identify subregions of high diagnostic value and, thus, accurately classify the whole slide. Other works combine the MIL approaches with well-known architectures of CNNs, such as ResNet [48,65,95], EfficientNetB1 [22], and SimCLR [59]. Moreover, Teramoto et al. [101] compared several CNNs as backbones (LeNet, AlexNet, ResNet, Inception, DenseNet) using MIL and an attention mechanism, while Hou et al. [57] presented 14 different combinations of expectation maximization (EM)-based MIL approach with Logistic Regression and Support Vector Machine (SVM). Finally, another work that attempted to overcome the lack of labeled data employs a semi-supervised approach inspired by YOLOv5 for the detection of micropapillary lung ADC. This method implements a teacher model, which is directly trained by the ground truth data, and a student model, which indirectly learns from the teacher model [68].

From a technical perspective, the extremely large size of images and the complexity of classification or detection problems in these images as well generate a very demanding process in terms of computational resources and training time of supervision. Typically, researchers can follow two main different approaches: (i) to develop a custom architecture, implementing all the components of both convolutional and fully connected layers and defining all the super parameters of the network or (ii) to use already pre-trained architectures and take advantage of transfer learning from other datasets (i.e., IMAGENET). Custom architectures can be more accurate than pre-trained CNNs with transfer learning if they are designed well for a specific problem and trained on an adequate set of images. However, they require more time and resources to develop and train. For these reasons, custom CNNs are mostly less deep than the pre-trained models to overcome the limitations of the demanding implementation and the computational requirements. Thus, most of the presented custom architectures for lung cancer consist of up to three convolution layers as well as up to three fully connected layers [21,49,61,74,82,97]. One of these works utilizes two different color spaces developing two same feature extractors, one for RGB and one based on HLS [82]. More extended architectures schemas have also been presented, developing six convolutions and two dense layers [84] or more than five convolution layers along with one devolution for upscaling [40,104]. Finally, the deeper CNN in this category, called Deep Hipo, operates on both magnifications (20× and 5×), and it is based on CAT-NET developing 19 layers in total [28].

As a result of the limited implementation effort needed, the vast majority of DL methods presented in the literature for lung cancer leverage well-known Convolutional Networks architectures, which are often pre-trained in different datasets. Several of these use architectures included in cloud-based platforms or frameworks, such as HALO-AI [30,78,81,89], HEAL Platform [50], AIFORIA [110], and Caffe [53,64,78]. In these cases, many well-known CNNs have been employed. For example, Wang et al. [50] used InceptionV3, ResNet50, VGG19, MobileNetV2, ShuffleNetV2 and MNASNET, while Yu et al. [53] employed AlexNet, GoogLeNet (InceptionV3), VGG-16 and ResNet-50. For classification problems, the most employed architectures seem to be the ResNet-based models, such as ResNet-18 [63,93,99], ResNet-50 [69,83], and ResNet-101 [36,100], as well as the Inception-based models [55,56,86]. Apart from these, U-Net [116,119], Xception [58,94,96], Hover-Net [76,83] and InceptionV3 [41] have also been used in several studies. In comparative studies or studies which use multiple architectures of CNNs, several other models have been presented, such as NASNetLarge [27], EfficientNet [26], SqueezeNet [39], etc. On the other hand, few approaches utilized known DL detectors for segmentation or quantitation purposes. Choi et al. [118] detected PD-L1 positive and PD-L1 negative tumor cells using Faster R-CNN, while Cheng et al. [117] for the same problem employed YOLO. Finally, Wang et al. [75] detected six different classes of cells segmenting the images with Mask R-CNN.

More sophisticated DL methods have been proposed, either modifying known architectures of CNNs or combining two differing CNN architectures and CNN architectures with classic ML techniques. Most of the modified architectures are based on ResNet. DeepRePath [72] is a novel CNN model based on ResNet-50 that operates on different magnifications building two CNNs, while a similar approach proposed by Sha et al. [113] developed two branches for the processing of small and large field-of-view features of PD-L1 classes. On the other hand, SE-ResNet-50 [38] focuses on the improvement of the activation function introducing CroRELU. Other novel modifications of known architectures are the KimiaNet22 based on DenseNet [44], the MR-EM-CNN, which extracts hierarchical multiscale features on an EM-CNN model [43], the DSC-VGG16, which provides a dual scale categorization of PD-L1 classes based on VGG16 [112], the WIFPS model [31] based on EfficientNet-B5, and the novel architecture proposed by Gonzales et al. [103], which utilizes three different stains. Finally, Rączkowski et al. [79] developed a novel architecture called ARA-CNN, which is inspired by both ResNet and DarkNet models.

By combining different CNN models or CNNs with classic ML techniques, researchers attempt to provide better performance in several categories of lung cancer problems. Combinations of different CNN models presented in the literature are (i) ResNet-50 with U-Net [111], (ii) EfficientNet with U-Net [77], and (iii) DeepLadV3 with Incepetion-ResNetV2 [71]. By combining DL and ML approaches, Wang et al. [42] introduced the LungDIG architecture, which employs an Inception-V3 model along with a classic multilayer perceptron. Two other approaches extract deep features utilizing the convolution layers of CNNs and then provide predictions using logistic regression [19,62]. SVM has also been used in cooperation with CNN models. Perez et al. [46] merged information from ResNet-18 from the processing of WSIs along with SVM from RNA-sequencing data, while Toğaçar et al. [45] and Hu et al. [88] combined SVM with DarkNet and Xception models, respectively. Finally, principal component analysis (PCA) techniques have been used along with CNNs architectures for dimensionality reduction of the extracted features [18,88]. The contribution of DL in lung cancer presents several other methods that employ Graph-based CNNs, GANs, and autoencoders. Graph CNNs have been used to identify regions or cell structural features that are highly associated with the class label. In this category, three approaches have been proposed, where Graph-based modules are combined with AlexNet [54], VGG16 [66], and ResNet. [37]. GANs are mostly used to generate informative synthetic sets of images in order to increase the training set and, thus, avoid overtraining issues. DASGAN, which is an extension of the CycleGAN architecture, has been introduced [114], merging two stains and leading deep survival learning methodology. Teramoto et al. introduced a progressive growing approach of GANs (PGGAN) combined with the VGG-16 model [98], while Mayer et al. [92] combined GANs with semi-supervised learning. Another auxiliary classifier GANs (AC-GANs) approach has been proposed by Kapil et al. [115] to generate classifier models and detect ALK and ROS1 fusions directly from H&E images. Finally, an unsupervised DL model that employs stacked autoencoders has been developed by Sheikh et al. [67].

## 4. Discussion

DL is progressively embraced in Pathology, especially for breast, colorectal, prostate, and lung cancer diagnosis, transforming the current landscape of medicine [120,121,122,123,124,125,126,127,128,129,130,131]. AI could play a pivotal role in the multidisciplinary approach to diagnosis and patient management. As already underlined above, in lung cancer, classification, accurate diagnosis and subtyping depend on distinct morphological features among cancer cells combined with staining patterns, tumor biological characteristics, and molecular data of mutations. Lung cancer histology is characterized by cellular heterogeneity, challenging the diagnostic process [132]. Several histological features can be defined by examining a single H&E-stained slide, such as glandular differentiation in lung ADC, the presence of keratinization and intercellular bridges in SCC, as well as scant cytoplasm and poorly defined cell borders in SCLC. However, for differential diagnosis, special immunohistochemical staining is required for accuracy. According to the WHO guidelines, the terminology for lung cancer classification in small biopsies or cytology and resection specimens must follow the proposed criteria [2]. For example, in resection specimens, lung ADC cases must be morphologically determined by the predominant histological pattern (lepidic, acinar, papillary, micropapillary, solid). The distinction of lung neuroendocrine tumors (NETs) directly from the H&E slide can also be challenging, whereas NETs are further classified as typical carcinoids, atypical carcinoids, SCLC, and LCNEC. Given that small biopsies and cytology specimens are encountered for diagnosis in about 70% of the patients, the available diagnostic material is often limited and thus, every effort should be employed to preserve sufficient material for molecular analysis. Therefore, it is strongly recommended to use only a limited panel of biomarkers, including the most representative ones for immunostaining for differential diagnosis. However, this approach can hamper accurate diagnosis. Here, AI could be of great help to the pathologist by guiding with high accuracy the prevailing diagnosis from an H&E-stained slide.

Data extraction of our systematic review demonstrated that DL-based methodologies for lung cancer diagnosis are mainly performed on histological H&E WSIs, with ADC versus SCC being the predominant classification task, as shown in Table 2. All the studies were performed with high classification accuracy for identifying ADC and SCC. Secondly, many studies utilized different CNN architectures for classifying ADC, SCC, and SCLC in small biopsies. The higher performance was in the study of Kanavati et al. [24] (AUC of 0.94–0.99), which included a large number of images. Only two studies designed a classification task for identifying ADC, SCC, SCLC, and LCNEC on WSIs [27,28]. This 4-class task represents the realistic daily practice of a pathologist. In both studies, the AUC was over 0.90, encouraging the fact that such DL models could be employed and of great value in a pathology laboratory. The third most common approach in histological slides was the employment of DL-based models for lung ADC histological subtyping. The studies of Sheikh [67] and DiPalma et al. [65] achieved the highest classification accuracy performing a 5-class problem (lepidic, acinar, papillary, micropapillary, solid). Albeit limited in number, eight noteworthy studies utilized cytological slides for lung cancer diagnosis or classification. Four of them performed a binary classification task for benign and malignant cell detection [98,99,100,101]. All studies showed good classification accuracy; however, compared to the classification problems performed on histological data, the dataset was limited in the majority of the studies. In addition, in the cytology section, the most common classification task for lung cancer (ADC, SCC, and SCLC) resulted in modest classification accuracies, including state-of-the-art architectures (66–77% and ~71%), with the main limitation being the small number of images included for training (55 and 76 cytological slides, respectively) [97,102]. Prediction of OS and risk of recurrence as well as identification of prognostic features, were also the aim of many research papers, in which the predicted output emerged after nuclei segmentation, TILs quantification, identification of gene expression, or clinical data. The highest AUC (0.917) for ALK rearrangements prediction was in the study by Chen et al. [31], while EGFR mutations were predicted with an AUC of 0.824, 0.84, and 0.83 in the studies by Wang, Yang, and Coudray et al., respectively [31,50,56,87]. In the most recent study by Pao et al., the prediction of EGFR mutational status in 2099 lung ADC tissue specimens reached an AUC of 0.87 [95]. As far as PD-L1 quantification is concerned, the majority of studies included datasets consisting of WSIs stained with the 22C3 antibody. The remaining studies included slides stained with the SP263 antibody or a combination of 22C3 and SP263 antibodies. For quantitative problems, such as TPS estimation for PD-L1 expression, labeling ground truth must be as consistent as possible to avoid misclassification concerning the specific cut-off points for PD-L1 evaluation DL-based models for PD-L1 TPS estimation offer several advantages to pathologists as TPS quantification is a time-consuming process prone to subjective estimation. Despite the extensive research and progress on histological images, further research on cytological material, including a larger dataset, is considered essential for optimizing classification performance.

According to the technical point of view, summarizing the methods presented in the literature, most of them (78 studies) developed supervised learning methodologies, basically dealing with classification problems of the medical question. Specifically, 11 studies implemented custom CNN architectures, 36 studies employed known models with or without transfer learning, 11 studies modified known architectures, and, finally, 14 studies combined CNNs either with each other or with ML techniques. Apart from the above crisp categories of supervised learning, the category named “other methods” contained six supervised, one weakly supervised, and one unsupervised method (eight studies in total). Weakly supervised methods are 13 in total, while there are one semi-supervised and one unsupervised method (Figure 2).

To conclude about the most commonly used known architectures, the employed architectures have been counted for each study, and the results are presented in Figure 3. Note that several studies have not used known architectures (for example the studies that develop custom CNN architectures), while several studies employ more than one.

Our review shows that many of the employed DL methods in lung cancer are particularly extensive and sophisticated, as well as scalarly evolving into new techniques following the development of AI. According to the comparative studies presented in this review, DL methods overall outperform traditional ML techniques. This superiority of DL could partially be explained by the quality of the features feeding the fully connected layers. The features in CNNs are not selected subjectively by the specialists but are automatically extracted from the convolutional layers, maximizing the carried information.

Comparing the reviewed architectures, it is evident from the results of the review that ResNet-based and Inception-based architectures have been used in about half of the methods presented in the literature, showing high performances compared to other architectures. The existence of residual blocks in most of these architectures (all ResNet and InceptionV4 models) seems to operate efficiently and effectively in biopsy image processing. Jumping features directly from a convolutional layer to many subsequent layers operates like merging features from different digital magnifications of scanning. Such a procedure seems to make sense for biopsy imaging, where different magnifications of scanning provide different knowledge about the microenvironment of the cells.

It is also meaningful to summarize the limitations of the DL techniques in lung cancer. Table 8 emphasizes several limitations of the application of the proposed DL methodologies in lung cancer diagnosis that we were able to identify based on our systematic review. Some of them are generally well-known constraints, while some others are related to the imaging problem of lung biopsies.

Our findings demonstrate that the field of Digital Pathology for lung cancer diagnosis has evolved rapidly in the last 5 years. However, at least for most laboratories, the use of these capabilities in daily clinical practice is still in its early stages. Adopting a fully digital workflow can be challenging, and limitations must be overcome for implementation in the clinical setting. Digital slide generation is the first step in moving from traditional to Digital Pathology. WSI scanners provide high-quality images of histological and cytological slides. These images can be uploaded and remotely reviewed by pathologists and cytologists on a computer, while they can be available for review by multiple pathologists. However, the organization and storage of large amounts of digitized data require high computing power, storage space, technical infrastructure, and backup capability. Furthermore, as a consequence of digitized data, ethical issues are arising concerning the sharing of sensitive personal data. DL models require large amounts of data for training, testing, and validation, which are retrieved from hospital archives. Therefore, a regulatory framework is essential to protect patient’s rights and ensure the security of sensitive medical data and confidentiality.

## 5. Conclusions

The field of Digital Pathology is evolving rapidly and, in the following years, is expected to be an inextricable part of a pathology laboratory. As highlighted above, AI-based approaches in Pathology are accompanied by several advantages, yet many challenges remain to be considered. Research for lung cancer diagnosis, prognosis, and prediction using DL methods is constantly improving to provide more accurate and reliable results. Moreover, for quantitative tasks, such as PD-L1 TPS estimation, the need for AI-based models is underlined because of their ability to provide reliable and objective assessment, eliminating subjective estimations that lead to intra- and inter-observer variability. The ongoing research and the efforts being made are at the forefront of transforming cancer diagnosis and treatment.

## Figures and Tables

**Figure 1 cancers-15-03981-f001:**
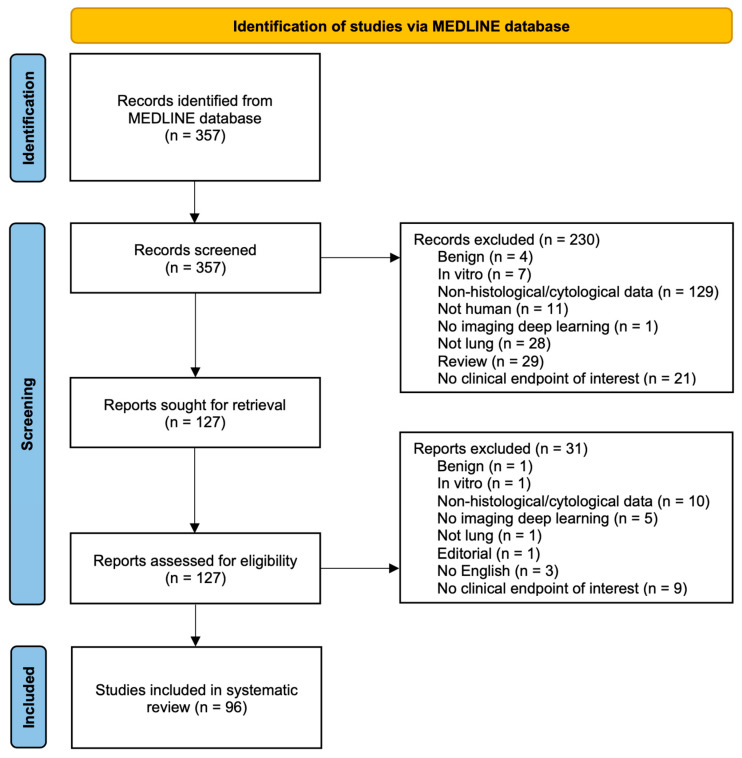
Flow diagram of the study selection process illustrating the systematic search and screening strategy along with the number of included and excluded studies.

**Figure 2 cancers-15-03981-f002:**
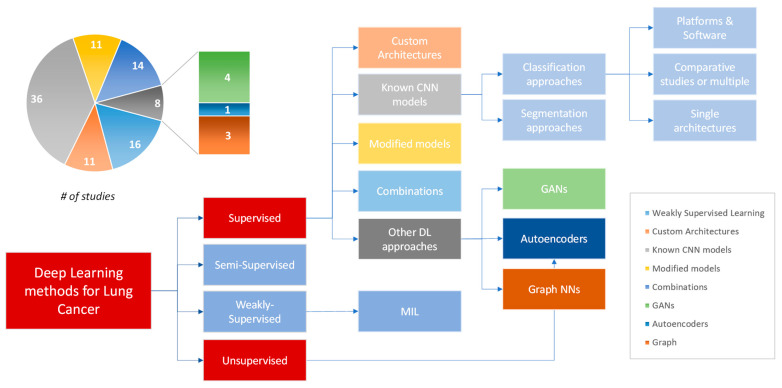
Summary of DL methods for lung cancer.

**Figure 3 cancers-15-03981-f003:**
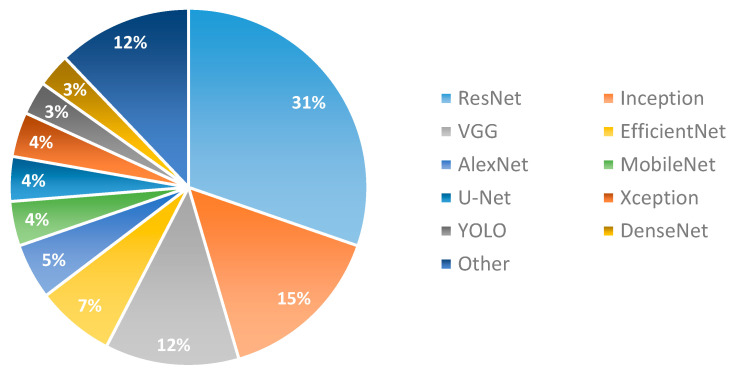
Distribution of known DL architectures for lung cancer.

**Table 1 cancers-15-03981-t001:** Characteristics of studies developing models for lung cancer diagnosis on histological images.

1st Author, Year	Technical Method	Classification	Dataset	Performance Metrics
Jain, 2022 [18]	Kernel PCA combined with Fast Deep Belief Neural Network	Binary: cancerous/normal cells	15,000 images from LZ2500 dataset, 215 tiles from the NLST dataset and 1634 images from NCI Genomic dataset	Acc: 97.10% in LZ2500 dataset, 98.00% in NLST dataset and 97.50% in NCI Genomic dataset
Civit-Masot, 2022 [23]	Custom Architecture with 3 Convolution and 2 dense layers	Binary: benign/malignant	15,000 images from LC25000 dataset	Overall Acc: 99.69% using 50 epochs
Tsuneki, 2022 [22]	Weakly supervised learning using EfficientNet-B1	Binary: ADC/non-ADC	8896 slides from Mita, Wajiro, Shinkuki, Shinkomonji and Shinmizumaki Hospitals	Acc: 85.30% Se: 88.50% Sp: 82.50%
Moranguinho, 2021 [21]	MIL approach using attention module and Grad-Cam algorithm	Binary: tumor/normal	3220 samples from TCGA dataset	*Standard Attention* Acc: 90.00% AUC: 0.94 *Gated Attention* Acc: 91.20% AUC: 0.95
Jiao, 2021 [19]	DELR: deep feature extraction and active learning for sample selection in Logistic Regression	Binary: tumor/non-tumor	338 ROIs from TCGA dataset	AUC: >0.95
Kanavati, 2020 [20]	Weakly supervised learning employing EfficientNet-B3 architecture	Binary: carcinoma/non-neoplastic	4204 WSIs from Kyushu Medical Center, 500 WSIs from International University of Health and Welfare, Mital Hospital, 680 WSIs from TCGA dataset and 500 WSIs from TCIA dataset	*Weakly supervised* AUC: 0.97–0.98 *Fully supervised* AUC: 0.88–0.96

Abbreviations: Acc, Accuracy; AUC, Area Under the Curve; DELR, Deep Embedding-based Logistic Regression; MIL, Multiple Instance Learning; NCI, National Cancer Institute; NLST, National Lung Screening Trial; PCA, Principal Component Analysis; ROI, Region Of Interest; Se, Sensitivity; Sp, Specificity; TCGA, The Cancer Genome Atlas; TCIA, The Cancer Imaging Archive; WSI, Whole Slide Image.

**Table 2 cancers-15-03981-t002:** Characteristics of studies developing models for lung cancer classification and non-small cell lung cancer subclassification on histological images.

1st Author, Year	Technical Method	Classification	Dataset	Performance Metrics
**Lung cancer classification**				
Yang, 2022 [27]	ResNet-152 VGG-19 Xception NASNetLarge	5-class: ADC/SCC/SCLC/ LCNEC/ non-tumor	205 WSIs from Gyeongsang National University Hospital	*Novel CNN model* Acc: 75.03% Macro-average AUC: 0.90
Chen, 2022 [31]	EfficientNet-B5 WSI-based IHC feature prediction system: a novel DL model based on EfficientNet-B5	Binary: normal/tumor tissue Binary: negative/positive expression of biomarkers 3-class: ADC/SCC/SCLC	1101 WSIs from First Affiliated Hospital of Sun Yat-sen University, Shenzhen People’s Hospital and Cancer Center of Guangzhou Medical University	*Tissue classification* Micro-average AUC: 0.98 Macro-average AUC: 0.99 *Biomarkers expression* AUC: 0.53–0.95 *3-class* Acc: 90.00%
Kosaraju, 2022 [28]	DEEP-HIPO: two magnifications (20× and 5×), based on CAT-NET with 19 layers	4-class: ADC/SCC/SCLC/ LCNEC	113 WSIs from Gyeongsang National University Hospital and 657 ADC WSIs from TCGA dataset	AUC: 0.96
Ilié, 2022 [30]	HALO-AI	4-class: SCLC/LCNEC/AC/ poorly differentiated ADC	150 NET and 25 poorly differentiated ADC WSIs from Laboratory of Clinical and Experimental Pathology of Nice University Hospital	Acc: 98.00% (95% CI: 93.70–1.00%) AUC: 0.93 F1-score: 0.99 (95% CI: 0.94–1.00)
Yang, 2021 [26]	EfficientNet-B5-based and ResNet-50-based DL model	6-class: ADC/SCC/SCLC/ pulmonary tuberculosis/organizing pneumonia/normal lung	1059 WSIs from First Affiliated Hospital of Sun Yat-sen University, 212 WSIs from Shenzhen People’s Hospital, and 422 WSIs from TCGA dataset	*EfficientNet-B5-based deep learning model* AUC: 0.97 in Sun Yat-sen University dataset 1, 0.92 in Sun Yat-sen University dataset 2, 0.96 in Shenzhen People’s Hospital dataset and 0.98 in TCGA dataset ICC: >0.87
Kanavati, 2021 [24]	Combination of EfficientNet-B1 and RNN	4-class: ADC/SCC/SCLC/ non-neoplastic	1723 WSIs from Kyushu Medical Center, 500 WSIs from Mita Hospital and 905 NSCLC WSIs from TCGA dataset	*Independent TBLB dataset of 83 indeterminate WSIs* AUC: 0.99 *1 independent TBLB and 3 independent surgical resection datasets of 2407 WSIs* AUC: 0.94–0.99
Wang, 2020 [25]	Modification of VGG-16	4-class: ADC/SCC/SCLC/ normal	939 WSIs from Sun Yat-sen University Cancer Center and 500 WSIs from TCGA dataset	Acc: 97.30% in Sun Yat-sen University Cancer Center dataset and 82.00% in TCGA dataset AUC: 0.86 in TCGA dataset
Kriegsmann, 2020 [29]	VGG-16 InceptionV3 InceptionResNetV2	4-class: ADC/SCC/SCLC/ skeletal muscle	270 cases from Institute of Pathology, University Clinic Heidelberg	*InceptionV3 with weights trained on the training dataset* Acc: 86.00% in validation dataset using 20 epochs and 85.00% in validation dataset using 50 epochs
**NSCLC subclassification**				
Mengash, 2023 [32]	MPADL-LC3 algorithm based on MobileNet and DBN	5-class: lung ADC/lung SCC/lung benign tissue/colon ADC/colon benign tissue	25,000 images from LC25000 dataset	*In testing phase using 80% of the dataset for training and 20% for testing* Acc: 99.42% for lung ADC, 99.28% for lung SCC and 99.30% for lung benign tissue
Al-Jabbar, 2023 [33]	ANN GooLeNet VGG-19	5-class: lung ADC/lung SCC/lung benign tissue/colon ADC/colon benign tissue	25,000 images from LC25000 dataset	*ANN with fusion features of VGG-19 and handcrafted * Acc: 99.60% for lung ADC, 99.80% for lung SCC and 99.70% for lung benign tissue
Wang, 2023 [34]	A novel multiplex-detection-based MIL model	Binary: ADC/SCC	993 WSIs from TCGA dataset	*Overall metrics* Acc: 90.52% AUC: 0.96
Patil, 2023 [35]	HistoROI: a ResNet18-based 6-class classifier	Binary: ADC/SCC	1034 WSIs from TCGA dataset	AUC: 0.93
El-Ghany, 2023 [36]	ResNet 101	5-class: lung ADC/lung SCC/lung benign tissue/colon ADC/colon benign tissue	25,000 images from LC25000 dataset	*Average overall metrics* Acc: 99.94% Sp: 99.96% Pr: 99.84% Re: 99.85% F1-score: 99.84%
Zheng, 2022 [37]	Graph-based modules with ResNet	3-class: ADC/SCC/normal	2071 WSIs from 435 patients from the CPTAC dataset, 2082 WSIs from 996 patients from TCGA dataset and 665 WSIs from 345 patients from NLST dataset	*Five-fold cross-validation * Acc: 91.20% ± 2.50% AUC: 0.98 *External test data* Acc: 82.30% ± 1.00% AUC: 0.93
Liu, 2022 [38]	SE-ResNet-50 with novel activation function CroRELU	3-class: infiltration/microinfiltration/normal 5-class: lung ADC/lung SCC/normal lung/colon ADC/normal colon	766 lung WSIs from First Hospital of Baiqiu’en and 25,000 images from LC25000 dataset	*3-class* Acc 98.33% *5-class* Acc: 99.96% Se: 99.86% Pr: 99.87%
Attallah, 2022 [39]	ShuffleNet, SqueezeNet, and MobileNet: 3 pre-trained lightweight CNN models	5-class: lung ADC/lung SCC/lung benign tissue/colon ADC/colon benign tissue	25,000 images from LC25000 dataset	Acc: 99.30% for lung ADC, 99.00% for lung SCC and 100.00% for lung benign tissue
Civit-Masot, 2022 [23]	Custom Architecture with 3 Convolution and 2 dense layers	3-class: ADC/SCC/benign	15,000 images from LC25000 dataset	*Colour CNN classifier* Overall Acc: 97.11% using 50 epochs *Greyscale CNN classifier* Overall Acc: 94.01% using 50 epochs
Wang, 2022 [40]	A custom architecture consisting of 5 Convolution and 3 Fully Connected layers along with a segmentation branch for up-sampling	3 class: ADC/SCC/normal	312 images from 36 patients from Qilu Hospital of Shandong University	DSC: 93.50% for segmenting SCC and 89.00% for segmenting ADC Acc: 97.80% in classifying SCC versus normal tissue and 100.00% in classifying ADC versus normal tissue
Dolezal, 2022 [37]	CNN models based on Xception architecture	Binary: ADC/SCC	941 WSIs from TCGA dataset, 1.306 from CPTAC dataset and 190 slides from Mayo Clinic dataset	AUROC: 0.96 at maximum dataset size for non-uncertainty quantification models AUROC: 0.98 at maximum dataset size for uncertainty quantification models
Le Page, 2021 [41]	A novel CNN model based on InceptionV3	Binary: squamous/non-squamous NSCLC	132 slides from Dijon University Hospital, 65 slides from Caen University Hospital, 60 slides from TCGA database and 1 cytological pericardium specimen	*Based on WSIs * Acc: 99.00% in the training dataset, 87.00% in validation dataset, 85.00% in the test dataset, 85.00% in the external validation cohort and 75.00% in TCGA dataset *Based on virtual TMAs * Acc: 99.00% in training dataset, 83.00% in validation dataset, 88.00% in test dataset, 92.00% in external validation cohort and 83.00% in TCGA dataset AUC: 0.94 in external validation cohort and 0.77 in TCGA dataset
Wang, 2021 [42]	LungDIG: Combination of InceptionV3 with multilayer perceptron	Binary: ADC/SCC	988 samples with both CNV and histological data	Acc: 87.10% AUC: 92.70% F1-Score: 87.60%
Zhao, 2021 [43]	MR-EM-CNN: Hierarchical multiscale features on EM-CNN	Binary: ROI/non-ROI Binary: ADC/SCC	2125 slides from TCGA dataset	*ROI localization* F1-score: 0.88 AUC: 0.96 *NSCLC classification* Se: 94.74% Sp: 85.83% F1-score: 0.90 AUC: 0.96
Dehkharghanian, 2021 [44]	KimiaNet-22: a DL model based on DenseNet	Binary: ADC/SCC	735 WSIs from TCGA dataset and 23 WSIs from Grand River Hospital	*Validation Sample* Pr: 92.00% Re: 91.00% F1-score: 0.91
Toğaçar, 2021 [45]	DarkNet-19 combined with YOLO and SVM	5-class: lung ADC/lung SCC/lung benign tissue/colon ADC/colon benign tissue	25,000 images from LC25000 dataset	Acc: 99.73% for lung ADC, 99.74% for lung SCC, 99.98% for lung benign tissue
Carrillo-Perez, 2021 [46]	Merging ResNet-18	3-class: ADC/SCC/healthy	1420 WSIs and 980 RNA-sequencing data from TCGA dataset	*Histology Classifier* Acc: 86.03% F1-Score: 83.39% AUC: 0.95
Lu, 2021 [47]	Clustering-Constrained-Attention MIL: a novel DL-based weakly supervised model	Binary: ADC/SCC	131 resection and 110 biopsy NSCLC WSIs from Brigham and Women’s Hospital, 993 NSCLC WSIs from TCGA dataset, and 974 NSCLC WSIs from TCIA dataset	*Public NSCLC WSI dataset (TCGA* and *TCIA*) AUC: 0.96 ± 0.02 using 100%, 0.95 ± 0.02 using 75% and 0.94 ± 0.02 using 50% of cases in training dataset *Independent NSCLC WSI dataset* (*Brigham and Women’s Hospital*) AUC: 0.94 ± 0.02 using 100%, 0.92 ± 0.01 using 75% and 0.88 ± 0.02 using 50% of cases in training dataset
Chen, 2021 [48]	MIL combined with ResNet-50	3-class: ADC/SCC/non-cancer	9662 WSIs from 2843 patients from Taipei Medical University Hospital, Taipei Municipal Wanfang Hospital and Taipei Medical University Shuang-Ho Hospital and 532 WSIs from TCGA dataset	AUC: 0.96 for ADC and 0.94 for SCC
Masud, 2021 [49]	Custom CNN architecture consisting of 3 Convolution and 1 Fully Connected layers	5-class: lung ADC/lung SCC/benign lung tissue/colon ADC/benign colonic tissue	25,000 images from LC25000 dataset	*Testing dataset * Acc: 96.33% Pr: 96.39% Re: 96.37% F1-score: 96.38%
Wang, 2021 [50]	InceptionV3, ResNet-50, VGG-19, MobileNetV2, ShuffleNetV2 and MNASNET on HEAL Platform	3-class: ADC/SCC/normal	NSCLC WSIs from TCGA dataset	AUC: 0.98 for ADC, 0.98 for SCC and 0.99 for normal
Kobayashi, 2020 [51]	A proposed modification to Diet Networks	Binary: ADC/SCC	950 patients from Pan-Lung Cancer dataset	Acc: ~80.00%
Xu, 2020 [52]	Hierarchical multiscale features on EM-CNN	Binary: tumor/normal Binary: ADC/SCC	2125 images from TCGA dataset	*Tumor/normal * AUC: 1.00 *ADC/SCC * AUC: 0.97
Yu, 2020 [53]	AlexNet GoogLeNet VGGNet-16 ResNet-50	Binary: ADC/benign Binary: SCC/benign Binary: ADC/SCC 3-class: terminal respiratory unit/proximal-inflammatory/proximal-proliferative ADC transcriptome subtype 4-class: classical/basal/secretor/primitive SCC transcriptome subtype	884 WSIs from TCGA dataset and 125 images from ICGC dataset	*ADC/benign* AUC: 0.95–0.97 in TCGA test dataset and 0.92–0.94 in ICGC test dataset *SCC/benign* AUC: 0.94–0.99 in TCGA test dataset and >0.97 in ICGC test dataset *ADC/SCC * AUC: 0.88–0.93 in TCGA test dataset and 0.73–0.86 in ICGC test dataset *ADC transcriptome subtype* AUC: 0.77–0.89 *SCC transcriptome subtype* AUC: ~0.70
Shi, 2019 [54]	Graph temporal ensembling: a novel semi-supervised CNN model based on AlexNet	Binary: ADC/SCC	2904 NSCLC image patches from WSIs of 42 patients from TCGA	Acc: 90.50% using 20% labeled patients, 91.00% using 35% labeled patients, 91.10% using 50% labeled patients and 94.00% using all labeled patients
Khosravi, 2018 [55]	CNN-basic InceptionV3-Last layer-4000 steps InceptionV3-Last layer-12,000 steps InceptionV1-Fine tune Inception-ResNetV2-Last layer InceptionV3-Fine tune	Binary: ADC/SCC	1273 images from TMAD and 3149 from TCGA dataset	*InceptionV1-Fine tune* Acc: 92% for TMAD images, 100% for TCGA intra-images and 83% for TCGA inter-images
Coudray, 2018 [56]	InceptionV3	Binary: tumor/normal 3-class: normal/ADC/SCC	1634 WSIs from Genetic Data Commons database and 340 slides from New York University Langone Medical Center	*Binary* AUC: 0.99 *3-class* AUC: 0.97
Hou, 2016 [57]	14 different combinations of EM-based MIL approach with CNN and multiclass logistic regression or SVM	3-class: ADC/SCC/ADC with mixed subtypes	718 WSIs from 641 patients from TCGA dataset	Acc: 79.80%

Abbreviations: AC, Atypical Carcinoid; Acc, Accuracy; ADC, Adenocarcinoma; ANN, Artificial Neural Network; AUC, Area Under the Curve; AUROC, Area Under the Receiver Operating Characteristic; CI, Confidence Interval; CNN, Convolutional Neural Network; CNV, Copy Number Variation; CPTAC, Clinical Proteomic Tumor Analysis Consortium; DSC, Dice Similarity Coefficient; EM, Expectation-Maximization; ICC, Interclass Correlation Coefficient; ICGC, International Cancer Genome Consortium; LCNEC, Large Cell Neuroendocrine Carcinoma; MIL, Multiple Instance Learning; MR, Multi-Resolution; NET, Neuroendocrine Tumor; NLST, National Lung Screening Trial; NSCLC, Non-Small Cell Lung Cancer; Pr, Precision; Re, Recall; RNA, Ribonucleic Acid; RNN, Recurrent Neural Network; ROI, Region Of Interest; SCC, Squamous Cell Carcinoma; SCLC, Small Cell Lung Cancer; Se, Sensitivity; Sp, Specificity; SVM, Support Vector Machine; TBLB, Transbronchial Lung Biopsy; TCGA, The Cancer Genome Atlas; TCIA, The Cancer Imaging Archive; TMA, Tissue Microarray; TMAD, Tissue Microarray Database; WSI, Whole Slide Image.

**Table 3 cancers-15-03981-t003:** Characteristics of studies developing models for the identification of lung adenocarcinoma predominant architectural pattern.

1st Author, Year	Technical Method	Classification	Dataset	Performance Metrics
Gao, 2022 [68]	Inspired by YOLOv5	Binary: micropapillary/non-micropapillary	ADC WSIs from Shandong Provincial Hospital	*Supervised model* Pr: 76.20% Re: 88.40% *Semi-supervised model* Pr: 77.50% Re: 89.60%
Xiao, 2022 [66]	GCNs combined with VGG-16	5-class: lepidic/acinar/papillary/ micropapillary/solid	243 images from 243 patients from Shandong Provincial Hospital	*LAD-GCN * Acc: 90.35% Pr: 86.53–98.34% Re: 85.80–98.78% F1-score: 0.86–0.99
Sheikh, 2022 [67]	Unsupervised deep learning model which employs stacked autoencoders	5-class: lepidic/acinar/papillary/ micropapillary/solid	31 WSIs from Dartmouth-Hitchcock Medical Center	Acc: 94.60% Se: 94.10% Pr: 94.20% F1-score: 0.94
Maleki, 2022 [69]	Four novel CNN models based on ResNet-50	Binary: solid/acinar	110 WSIs from Dartmouth-Hitchcock Medical Center	Acc: 65.90–99.90%
Sadhwani, 2021 [62]	InceptionV3 and Deep features extraction combined with logistic regression in two stages	9-class: acinar/lepidic/solid/papillary/micropapillary cribriform/necrosis/leukocyte aggregates/other	ADC WSIs from TCGA dataset and 50 ADC WSIs for external validation from an independent pathology laboratory in the United States	AUC: 0.93 in TCGA dataset and 0.92 in external validation dataset
DiPalma, 2021 [65]	MIL approach using ResNet	5-class: lepidic/acinar/papillary/ micropapillary/solid	269 slides from TCGA dataset and Dartmouth-Hitchcock Medical Center	Acc: 94.51% (95% CI: 92.77–96.20%) Pr: 80.41% (95% CI: 70.55–89.56%) Re: 81.67% (95% CI: 71.20–90.43%) F1-score 0.80 (95% CI: 0.71–0.88)
Wei, 2019 [63]	ResNet-18	6-class: lepidic/acinar/papillary/ micropapillary/solid/benign	422 ADC WSIs from Dartmouth-Hitchcock Medical Center	AUC: 0.97–1.00
Gertych, 2019 [64]	GoogLeNet, ResNet-50 and modified AlexNet developed in Caffe engine	5-class: solid/micropapillary/ acinar/cribriform/non-tumor	50 cases from Cedars-Sinai Medical Center in Los Angeles, 33 cases from Military Institute of Medicine in Warsaw and 27 cases from TCGA dataset	Overall Acc: 89.24%

Abbreviations: Acc, Accuracy; ADC, Adenocarcinoma; AUC, Area Under the Curve; CI, Confidence Interval; CNN, Convolutional Neural Network; GCN, Graph Convolutional Network; MIL, Multiple Instance Learning; Pr, Precision; Re, Recall; Se, Sensitivity; TCGA, The Cancer Genome Atlas; WSI, Whole Slide Image.

**Table 4 cancers-15-03981-t004:** Characteristics of studies developing models for the prediction of lung cancer prognosis using histological data.

1st Author, Year	Aim of Study	Technical Method	Classification	Dataset	Performance Metrics
Liu, 2023 [60]	ADC prognosis prediction	MIM (MLP IN MLP): a novel deep learning-based model	3-class: infiltration/microinfiltration/ normal	780 images from the First Hospital of Jilin University	*Overall metrics in the test dataset * Acc: 95.31% Se: 93.10% Sp: 96.43% F1-score: 93.10% Pr: 93.09%
Yu, 2023 [85]	ADC prognosis prediction	Transformer-guided MIL with both handcrafted and deep features	Binary: negative/positive aneuploidy	Slides from 339 patients from TCGA dataset	*In lung ADC test dataset* Acc: 77.60% F1-score: 79.50% Cohen’s kappa: 0.55 AUC: 0.82
Qaiser, 2022 [80]	Lung cancer prognosis prediction	ResNet-18 along with attention mechanism	Binary: high/low OS	1122 WSIs from 410 patients from NLST dataset	C-index: 0.70
Shvetsov, 2022 [76]	NSCLC prognosis prediction	HoVer-Net	Binary: high-TIL/low-TIL	WSIs from CoNSeP, PanNuke, MoNuSAC and UiT-TILs datasets	*HoVer-Net PanNuke Aug model * HR: 0.30 (95% CI: 0.15–0.60) *HoVer-Net MoNuSAC Aug model* HR: 0.27 (95% CI: 0.14–0.53)
Guo, 2021 [77]	NSCLC prognosis prediction	EfficientUnet: a combination of EfficientNet and Unet ResNet	Binary: tumor/non-tumor area Binary: positive/negative tumor cell staining Binary: positive/negative TILs staining	1859 NSCLC TMAs from Medical University of Gdansk and 214 NSCLC WSIs from Shanghai Pulmonary Hospital	*Integrated score in the training dataset* AUC: 0.90 for OS and 0.85 for RFS *Res-score in the external validation dataset* AUC: 0.80–0.87 for OS and 0.83–0.94 for RFS
Pan, 2022 [83]	ADC prognosis prediction	ResNet-50 HoVer-Net	Binary: high-risk/low-risk	Patients from Guangdong Provincial People’s Hospital, Shanxi Cancer Hospital, Yunnan Cancer Hospital and TCGA	*In terms of OS * HR: 2.68 in discovery cohort, 3.05 in validation cohort 1, 2.39 in validation cohort 2 and 1.99 in validation cohort 3 *In terms of DFS* HR: 2.07 in discovery cohort, 1.54 in validation cohort 1, and 3.80 in validation cohort 2
Levy-Jurgenson, 2020 [86]	ADC prognosis prediction	5 deep learning models based on InceptionV3	Binary: low/high heterogeneity index	469 ADC slides from TCGA dataset and mRNA/miRNA expression data from GDC database	Log rank *p*-value: 0.07
Wang, 2020 [75]	ADC prognosis prediction	Mask-RCNN	Binary: high-risk/low-risk	208 images from 135 patients from NLST dataset and 431 histological images from 372 patients from TCGA dataset	HR: 2.23 (95% CI: 1.37–3.65)
Wang, 2019 [74]	ADC prognosis prediction	ConvPath: A custom architecture with 2 convolution layers	Binary: high-risk/low-risk	1337 images from 523 patients from TCGA dataset, 345 images from 201 patients from NLST dataset, 102 images from 102 patients from Chinese Academy of Medical Sciences dataset and 130 images from 112 patients from Special Program of Research Excellence dataset	Log rank *p*-value: <0.01 in TCGA dataset and 0.03 in Chinese Academy of Medical Sciences dataset
Wu, 2020 [73]	Lung cancer recurrence and metastasis prediction	DeepLRHE: a novel deep learning model consisting of a CNN and a ResNet component	Binary: high-risk/low-risk	211 images from TCGA dataset	Se: 84.00% Sp: 67.00% Pr: 78.00% F1-score: 81.00% AUC: 0.79
Hattori, 2022 [82]	ADC recurrence prediction	Custom Architecture consisting of 3 Convolution and 1 Fully Connected layer in different color spaces	Binary: presence/absence of recurrence	WSIs from 55 stage IB ADC patients	Se: 91.70% Sp: 90.20% Acc: 90.90%
Shim, 2021 [72]	ADC recurrence prediction	DeepRePath: a novel CNN model based on ResNet-50	Binary: high/low probability of recurrence within 3 years	3923 slides from 5 St. Mary’s hospitals affiliated with the Catholic University of Korea in Seoul, Incheon, Uijeongbu, Bucheon, and Yeouido and 1067 WSIs from TCGA dataset	HR: 5.56
Yang, 2021 [87]	Lung cancer immunotherapy efficacy prediction	DeepLRHE: a novel deep learning model consisting of a CNN and a ResNet component	Binary: positive/negative expression of TP53, EGFR, DNMT3A, PBRM1 and STK11	180 WSIs from TCGA dataset	AUC: 0.87 for TP53, 0.84 for EGFR, 0.78 for DNMT3A, 0.75 for PBRM1 and 0.71 for STK11
Barmpoutis, 2021 [71]	Lung cancer TLS identification and quantification	Combination of DeepLadV3 with Inception-ResNetV2	Binary: TLS/non-TLS region	Slides from 18 patients from Norfolk and Norwich University Hospital	Sp: 92.87% with Se: 95.00% Sp: 88.79% with Se: 98.00% Sp: 84.32% with Se: 99.00% AUROC: 0.96
Hu, 2021 [88]	Anti-PD-L1 response prediction	Combination of Xception, PCA, and SVM	Binary: response/non-response	190 melanoma slides from TCGA-SKCM dataset and 55 NSCLC slides from Guangdong Province Cancer Hospital	AUC: 0.65 (95% CI: 49.40–78.40%)

Abbreviations: Acc, Accuracy; ADC, Adenocarcinoma; AUC, Area Under the Curve; AUROC, Area Under the Receiver Operating Characteristic; CI, Confidence Interval; CNN, Convolutional Neural Network; DFS, Disease-Free Survival; GDC, Genomic Data Commons; HR, Hazard Ratio; miRNA, micro-Ribonucleic Acid; mRNA, messenger Ribonucleic Acid; NLST, National Lung Screening Trial; NSCLC, Non-Small Cell Lung Cancer; OS, Overall Survival; Pr, Precision; RCNN, Regional-Convolutional Neural Network; RFS, Relapse-Free Survival; Se, Sensitivity; Sp, Specificity; TCGA, The Cancer Genome Atlas; TIL, Tumor-Infiltrating Lymphocyte; TLS, Tertiary Lymphoid Structures; TMA, Tissue Microarray; WSI, Whole Slide Image.

**Table 5 cancers-15-03981-t005:** Characteristics of studies developing models for the prediction of lung cancer mutational status using histological data.

1st Author, Year	Technical Method	Classification	Dataset	Performance Metrics
Pao, 2023 [95]	An attention-based MIL model based on ResNet50	Binary: mutated/wild-type EGFR	2099 specimens	AUC: 0.87 NPV: 95.40% PPV: 41.00%
Dammak, 2023 [94]	VGG16 Xception NASNet-Large	Binary: high/low TMB	50 slides from TCGA dataset	*Per-patient metrics for the optimal model* (*VGG16*) AUC: 0.65 Acc: 65.00% Se: 77.00% Sp: 43.00%
Mayer, 2022 [92]	GANs along with unsupervised and semi-supervised learning	Binary: positive/negative ALK and ROS1 rearrangement	Slides from 234 advanced-stage NSCLC patients from Sheba Medical Center	Se: 100% for both ALK and ROS1 Sp: 100% for ALK and 98.57% for ROS1 NPV: 100% for both ALK and ROS1 PPV: 100% for ALK and 50.50% for ROS1
Terada, 2022 [89]	DenseNet via the HALO-AI platform	Binary: positive/negative ALK rearrangement	300 patients from Shizuoka Cancer Center, Shizuoka, Japan	*With 50% probability threshold* AUC: 0.73 (95% CI: 0.65–0.82) Acc: 73.00% Se: 73.00% Sp: 73.00% PPV: 73.00% NPV: 73.00% F1-score: 37.00%
Tomita, 2022 [90]	ResNet-18, EfficientNet-B0	Binary: mutated/wild-type BRAF, EGFR, KRAS, STK11, and TP53	747 WSIs from 232 patients from Dartmouth-Hitchcock Medical Center and 111 cases from CPTAC-3 study	*Internal test dataset from Dartmouth-Hitchcock Medical Center* AUC: 0.80 (95% CI: 0.69–0.90) for EGFR and 0.71 (95% CI: 0.61–0.81) for TP53 *External test dataset from CPTAC-3 study * AUC: 0.69 (95% CI: 0.62–0.75) for EGFR and 0.68 (95% CI: 0.60–0.75) for TP53
Rączkowski, 2022 [79]	ARA-CNN inspired by ResNet and DarkNet	Binary: mutated/wild-type ALK, BRAF, DDR2, EGFR, KEAP1, KRAS, MET, PIK3CA, RET, ROS1, STK11, TP53 and PDGFRB	Samples from 55 tumors from the Medical University of Lublin, Poland, and 467 images from TCGA dataset	AUC: up to 0.74 for PDGFRB
Niu, 2022 [93]	ResNet-18	Binary: high/low TMB	427 WSIs from 427 patients from TCGA dataset	AUC: 0.64
Li, 2022 [96]	Fine-tuned pre-trained Xception model	Binary: mutated/wild-type STK11, TP53, LRP1B, NF1, FAT1, FAT4, KEAP1, EGFR and KRAS	100,000 images from NCT-CRC-100k dataset and 900 ADC WSIs from TCGA dataset	AUC
Wang, 2021 [50]	InceptionV3, ResNet-50, VGG-19, MobileNetV2, ShuffleNetV2 and MNASNET on HEAL Platform	Binary: mutated/wild-type STK11, KEAP1, NF1, TP53, EGFR, FAT1, FAT4, LRP1B, SETBP1 and KRAS	NSCLC WSIs from TCGA dataset	AUC: 0.63 for STK11, 0.77 for KEAP1, 0.70 for NF1, 0.72 for TP53, 0.82 for EGFR, 0.55 for FAT1, 0.69 for FAT4, 0.76 for LRP1B, 0.54 for SETBP1, 0.66 for KRAS
Huang, 2021 [91]	DeepIMLH: a novel CNN model based on ResNet concept	Binary: mutated/wild-type AKT1, FGFR1, FGFR2, HRAS and MET	180 WSIs from TCGA dataset	Acc: 72.00% for AKT1, 83.00% for FGFR1, 82.00% for FGFR2, 79.00% for HRAS and 86.00% for MET AUC: 0.83 for FGFR1, 0.82 for FGFR2, 0.79 for HRAS and 0.86 for MET
Sadhwani, 2021 [62]	InceptionV3 and Deep features extraction combined with logistic regression in two stages	Binary: low/high TMB	ADC WSIs from TCGA dataset and 50 ADC WSIs for external validation from an independent pathology laboratory in the United States	AUC: 0.71 (95% CI: 0.63–0.79)
Coudray, 2018 [56]	InceptionV3	Binary: mutated/wild-type NF1, FAT4, LRP1B, KEAP1, KRAS, FAT1, TP53, SETB1, EGFR and STK11	1634 WSIs from Genetic Data Commons database and 340 slides from New York University Langone Medical Center	AUC: 0.64 for NF1, 0.64 for FAT4, 0.66 for LRP1B, 0.68 for KEAP1, 0.73 for KRAS, 0.75 for FAT1, 0.76 for TP53, 0.78 for SETB1, 0.83 for EGFR and 0.86 for STK11

Abbreviations: Acc, Accuracy; ADC, Adenocarcinoma; AUC, Area Under the Curve; CI, Confidence Interval; CNN, Convolutional Neural Network; CPTAC, Clinical Proteomic Tumor Analysis Consortium; GAN, Generative Adversarial Network; MIL, Multiple Instance Learning; NPV, Negative Predictive Value; Non-Small Cell Lung Cancer; PPV, Positive Predictive Value; Se, Sensitivity; Sp, Specificity; TCGA, The Cancer Genome Atlas; TMB, Tumor Mutation Burden; WSI, Whole Slide Image.

**Table 6 cancers-15-03981-t006:** Characteristics of studies developing models for the cytological interpretation of lung cancer.

1st Author, Year	Technical Method	Classification	Dataset	Performance Metrics
Tsukamoto, 2022 [102]	AlexNet GoogLeNet/InceptionV3 VGG-16 ResNet-50	3-class: ADC/SCC/SCLC	82 images from 36 ADC cases, 125 images from 14 SCC cases and 91 images from 5 SCLC cases	*AlexNet* Acc: 73.70% *GoogLeNet/InceptionV3* Acc: 66.80% *VGG16* Acc: 76.80% *ResNet50* Acc: 74.00%
Wang, 2022 [104]	Custom Architecture with 8 Convolution and 1 Deconvolution layers	Binary: positive/negative lymph node metastasis	122 WSIs from EBUS-guided TBNA samples from Tri-Service General Hospital	*Novel DL model* Pr: 93.40% in 1st and 91.80% in 2nd experiment Se: 89.80% in 1st and 96.30% in 2nd experiment DSC: 82.20% in 1st and 94.00% in 2nd experiment IoU: 83.20% in 1st and 88.70% in 2nd experiment
Xie, 2022 [99]	ResNet-18	Binary: benign/malignant	404 WSIs from Shangai Pulmonary Hospital	Acc: 91.67% Sp: 94.44% Se: 87.50% AUC: 0.95 (95% CI: 0.90–0.99)
Lin, 2021 [100]	ResNet-101	Binary: benign/malignant	499 images from 97 patients from National Taiwan University Cancer Center and National Taiwan University Hsin-Chu Hospital	Acc: 98.80% for patch-based classification, 95.50% for image-based classification and 92.90% for patient-based classification Se: 98.80% for patch-based classification Sp: 98.80% for patch-based classification
Teramoto, 2021 [101]	MIL approach with attention mechanism and several CNN architectures as backbone	Binary: benign/malignant	Images from 322 patients	Acc: 91.60%
Teramoto, 2020 [98]	Combination of progressive growing GAN and VGG-16 architecture	Binary: benign/malignant	Images from 60 patients	Acc: 85.30%
Gonzalez, 2020 [103]	A deep learning model based on InceptionV3	Binary: LCNEC/SCLC	114 cytological and histological slides from 40 cases	*Diff-Quik^®^-stained model* AUC: 1.00 with a threshold at Se: 100.00% and Sp: 87.50% *Pap-stained model* AUC: 1.00 with a threshold at Se: 100.00% and Sp: 85.70% *H&E-stained model* AUC: 0.88 with a threshold at Se: 100.00% and Sp: 87.50%
Teramoto, 2017 [97]	Custom architecture consisting of 3 convolutions and 3 Fully Connected layers	3-class: ADC/SCC/SCLC	76 cases	*Original images* Acc: 73.20% for ADC, 44.80% for SCC, 75.80% for SCLC and 62.10% overall *Augmented images* Acc: 89.00% for ADC, 60.00% for SCC, 70.30% for SCLC and 71.10% overall

Abbreviations: Acc, Accuracy; ADC, Adenocarcinoma; AUC, Area Under the Curve; CI, Confidence Interval; CNN, Convolutional Neural Network; DSC, Dice Similarity Coefficient; EBUS, Endobronchial Ultrasound; GAN, Generative Adversarial Network; H&E, Hematoxylin & Eosin; IoU, Intersection over Union; LCNEC, Large Cell Neuroendocrine Carcinoma; MIL, Multiple Instance Learning; Pr, Precision; SCC, Squamous Cell Carcinoma; SCLC, Small Cell Lung Cancer; Sp, Specificity; Se, Sensitivity; TBNA, Transbronchial Needle Aspiration; WSI, Whole Slide Image.

**Table 7 cancers-15-03981-t007:** Characteristics of studies developing models for the assessment of programmed cell death ligand 1 expression in lung cancer using histological data.

1st Author, Year	Technical Method	Classification	IHC Assay	Dataset	Performance Metrics
Cheng, 2022 [117]	MobileNetV2 for classification and YOLO for detection	3-class: PD-L1+ tumor cells/PD-L1+ immune cells/PD-L1− tumor cells	22C3 pharmDx (DAKO) and SP263 (Ventana)	1288 samples from Zhejiang Cancer Hospital	*Best model* LCC 95% CI: 0.86–0.89 with PD-L1 (22C3) assay and 0.81–0.91 with PD-L1 (SP263) assay
Choi, 2022 [118]	Faster R-CNN	Binary: PD-L1+/PD-L1− tumor cells	22C3 pharmDx (DAKO)	348 slides from Samsung Medical Center and 131 slides from Seoul National University Bundang Hospital	AUROC: 0.89 for PD-L1+ cells and 0.81 for PD-L1− cells F1-score: 72.30% for PD-L1+ cells and 72.20% for PD-L1− cells
Huang, 2022 [119]	U-Net based architecture	3-class: negative PD-L1 expression (TPS: <1%)/low PD-L1 expression (TPS: 1–49%)/high PD-L1 expression (TPS: ≥50%)	22C3 pharmDx (DAKO)	222 WSIs from Fudan University Shanghai Cancer Center	r_s_: 0.87 Acc: 79.13% for all subsets, 85.29% for negative TPS subset, 77.79% for low TPS subset, and 72.73% for high TPS subset
Hondelink, 2022 [110]	A novel supervised deep learning model based on AIFORIA CREATE software (v4.6)	3-class: TPS < 1%/1–49%/50–100%	22C3 pharmDx (DAKO)	199 stage IV NSCLC WSIs stained with PD-L1 22C3 antibody from Leiden University Medical Centre	ICC: 0.96 (95% CI: 0.94–0.97) Cohen’s kappa: 0.68
Wu, 2022 [116]	A novel supervised deep learning algorithm based on U-Net	Binary: tumor/non-tumor 3-class: TPS < 1%/1–49%/50–100%	22C3 pharmDx (DAKO) and SP263 (Ventana)	501 NSCLC WSIs from Peking University Cancer Hospital and Tianjin Medical University Cancer Hospital	*Binary* Acc: 93.26% Sp: 96.41% Pr: 92.48% Re: 86.09% F1-score: 88.71% IoU: 80.51% *3-class* r: 0.94–0.95 in 22C3 assay and 0.98 in SP263 assay
Kapil, 2021 [114]	DASGAN network: an extension of CycleGAN architecture An extension of the deep survival learning methodology	Binary: epithelial/non-epithelial 3-class: tumor PD-L1+ epithelial region/tumor PD-L1− epithelial region/other regions (immune, stromal, necrotic)	SP263 (Ventana)	56 WSIs stained with Pan-Cytokeratin and 122 WSIs stained with PD-L1 SP263 antibody	*Binary* F1-score: 88.60% *3-class* F1-score: 85.00%
Wang, 2021 [112]	DSC-VGG-16: a novel dual-scale categorization-based deep learning model based on VGG-16	4-class: PD-L1+ tumor cells/PD-L1− tumor cells/PD-L1+ immune cells/other region 3-class: maximum counts of PD-L1+ tumor cell (TP1)/50% PD-L1+ tumor cell of TP1 (TP2)/25% PD-L1+ tumor cell of TP1 (TP3) 3-class: TPS < 1%/1–49%/50–100%	22C3 pharmDx (DAKO)	300 NSCLC slides stained with PD-L1 22C3 antibody from Changhai and Changzheng hospitals	*TPS prediction* F1-score: 90.24% with 1% and 81.82% with 50% cut-off AUC: 0.97 with 1% and 0.99 with 50% cut-off Se: 88.10% with 1% and 75.00% with 50% cut-off Sp: 95.59% with 1% and 98.98% with 50% cut-off Cohen’s kappa: 0.79 (95% CI: 0.68–0.90) Lcc: 0.88 (95% CI: 0.83–0.92)
Liu, 2021 [111]	Automated Tumor Proportion Scoring System: a novel deep learning model using Res50UNet for tumor region segmentation and MicroNet for tumor nuclei detection	3-class: TPS < 1%/1–49%/50–100%	22C3 pharmDx (DAKO)	96 SCC WSIs stained with PD-L1 22C3 antibody from Fudan University Shanghai Cancer Center	Acc: 74.51% MAE: 8.65 (95% CI: 6.42–10.90) r: 0.94
Sha, 2019 [113]	Modified ResNet-18	3-class: tumor PD-L1+/tumor PD-L1−/other	22C3 pharmDx (DAKO)	130 NSCLC samples	AUC: 0.80 for all cases, 0.83 for ADC cases and 0.64 for SCC cases
Kapil, 2018 [115]	Auxiliary Classifier GAN	Binary: PD-L1+/PD-L1− tumor regions	SP263 (Ventana)	270 NSCLC slides from NCT01693562 and NCT02000947 clinical trials	Lcc: 0.94 r: 0.95 MAE: 8.00 OPA: 0.88 NPA: 0.90 PPA: 0.85

Abbreviations: Acc, Accuracy; ADC, Adenocarcinoma; AUC, Area Under the Curve; AUROC, Area Under the Receiver Operating Characteristic; CI, Confidence Interval; CNN, Convolutional Neural Network; GAN, Generative Adversarial Network; ICC, Interclass Correlation Coefficient; IHC, Immunohistochemistry; IoU, Intersection over Union; LCC, Linear Correlation Coefficient; Lcc, Lin’s concordance coefficient; MAE, Mean Absolute Error; NPA, Negative Percent Agreement; NSCLC, Non-Small Cell Lung Cancer; OPA, Overall Percent Agreement; PD-L1, Programmed cell Death Ligand 1; PPA, Positive Percent Agreement; Pr, Precision; r, Pearson’s correlation coefficient; Re, Recall; r_s_, Spearman’s rank correlation coefficient; SCC, Squamous Cell Carcinoma; Se, Sensitivity; Sp, Specificity; TPS, Tumor Proportion Score; WSI, Whole Slide Image.

**Table 8 cancers-15-03981-t008:** Deep learning limitations for lung cancer applications.

Limitation	Property
Lack of interpretability and explainability	According to the review, only a few approaches focus on performing tasks that require common sense reasoning, such as understanding the physical characteristics of the cells. More explainable artificial intelligence approaches could be proposed in the future.
Training limitations with inadequate samples	Deep learning algorithms require massive amounts of labeled data to achieve good performance, and thus, thousands of annotations must be performed by pathologists.
Less powerful in problems beyond classification	Deep learning algorithms are mainly designed for classification problems, such as image recognition and natural language processing. They are less effective for other types of problems, such as regression, clustering, etc.
Lack of global generalization	Deep learning algorithms often overfit the training data and fail to generalize to new or unlabeled data. For example, a deep learning model may perform well on images from a specific microscopic scanner but poorly on images from a different microscope.
High memory and computational cost requirements	The training of deep models using extremely large size of images, such as biopsies, constitutes a very demanding process in terms of computational resources and training time of the supervision.

## Data Availability

All data that support the findings of this study are available from the corresponding author upon reasonable request.

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
