# Peer review of "Deep Learning for Lung Cancer Diagnosis, Prognosis and Prediction Using Histological and Cytological Images: A Systematic Review"

_cancers, 2023, doi:10.3390/cancers15153981_

Round 1

Reviewer 1 Report

  1. Another major issue is the lack of rigorous methodology. The paper does not provide any information regarding the systematic search and selection of relevant articles or studies. A comprehensive review should demonstrate a well-defined methodology to ensure the inclusion of relevant and reliable sources.

  2. The review paper also suffers from a lack of critical analysis and evaluation of the reviewed technologies. Merely describing and summarizing the existing technologies without offering any meaningful insights or comparisons diminishes the value of the review. The paper should have critically examined the strengths, limitations, and potential areas of improvement for each technology. I suggest the authors read studies performed by scholars such as R. Ranjbarzadeh et al., s. Jafarzadeh et al., and their groups

  3. The overall organization and structure of the paper are inadequate. The flow of ideas is unclear, and there is a lack of coherence between sections. The paper should have presented a clear introduction, outlined the main themes or categories of technologies, and provided a concise summary or conclusion to tie the information together.
  4. Furthermore, the paper lacks depth in terms of discussing the ethical considerations and potential risks associated with user-centred AI-based healthcare technologies. It is essential to address the ethical implications of such technologies, including issues of privacy, data security, and bias.

Author Response

We thank reviewer 1 for their comments. Underneath, we present the changes made to our manuscript according to their suggestions.

1) Another major issue is the lack of rigorous methodology. The paper does not provide any information regarding the systematic search and selection of relevant articles or studies. A comprehensive review should demonstrate a well-defined methodology to ensure the inclusion of relevant and reliable sources.

Response: The revised manuscript now includes a detailed methodology section (lines 97-129).

2) The review paper also suffers from a lack of critical analysis and evaluation of the reviewed technologies. Merely describing and summarizing the existing technologies without offering any meaningful insights or comparisons diminishes the value of the review. The paper should have critically examined the strengths, limitations, and potential areas of improvement for each technology. I suggest the authors read studies performed by scholars such as R. Ranjbarzadeh et al., s. Jafarzadeh et al., and their groups.

Response: It is indeed very important to be able to interpret the performance of each CNN architecture and the comparative performance between of them. We included three paragraphs in the discussion section (Lines 756-778), in which we attempt to explain several aspects of the reviewed technologies, as well as to provide some meaningful insights or qualitative comparisons. We have also added Table 8, which emphasizes several limitations of deep learning regarding the proposed methodologies for lung cancer.

3) The overall organization and structure of the paper are inadequate. The flow of ideas is unclear, and there is a lack of coherence between sections. The paper should have presented a clear introduction, outlined the main themes or categories of technologies, and provided a concise summary or conclusion to tie the information together.

Response: The coherence between the paragraphs was improved by reading thoroughly the paper. Changes have been made in the presentation order of some studies that shared common either methodologies or medical questions. A paragraph was added to the introduction section and more information was added to the discussion section. The changes are highlighted in the text.

4) Furthermore, the paper lacks depth in terms of discussing the ethical considerations and potential risks associated with user-centered AI-based healthcare technologies. It is essential to address the ethical implications of such technologies, including issues of privacy, data security, and bias.

Response: We added new information about the ethical issues associated with the use of user-centered AI-based healthcare technologies in the discussion section (lines 779-793). The changes are also highlighted in the text.

Reviewer 2 Report

There are following points to be included in revised manuscript

1. Authors have to include the their contribution in the introduction section.

2. The manuscript is review article, so authors have to add the challenges and their interpretation of the study.

3. Minor language editing required.

4. Include few latest references.

5. Figure in not clearly visible, so add revised figure in updated manuscript.

6. Conclusion section is not making any sense, so rewrite it effectively.

Minor proof read required.

Author Response

We thank the reviewer 2 for the comments. Underneath, we present the changes made to our manuscript according to the proposed suggestions.

1) Authors have to include their contribution in the introduction section.

Response: We clearly stated our contribution as well as the main purpose of our study in the last paragraph of the introduction section (lines 75-92). The changes are also highlighted in the text.

2) The manuscript is review article, so authors have to add the challenges and their interpretation of the study.

Response: We have structured our paper so as to be informative for both pathologists and cytologists by providing a detailed analysis and a comprehensive guide of the existing DL applications for lung cancer and offering valuable information to researchers for further study. A paragraph was added to the introduction section and more information was added to the discussion section (lines 75-92). The changes are also highlighted in the text.

3) Minor language editing required.

Response: The whole manuscript underwent English revisions, and its language is now improved.

4) Include few latest references.

Response: We updated our systematic review by extending our search in PubMed until 31 March 2023 (lines 98-100). This procedure identified 25 new articles published in 2023, 9 of which fulfilled our criteria of eligibility and were finally included in our manuscript. As a result, the flow diagram (Figure 1) of our paper has been updated (lines 141-143). Information from the newly included articles has been added to the tables and main text of the revised manuscript (References 32-36, 61, 86, 95,96).

5) Figure in not clearly visible, so add revised figure in updated manuscript.

Response: The quality of the figures in the revised manuscript has been improved, while Figure 2 has been extended to the whole width of the manuscript to become more distinct.

6) Conclusion section is not making any sense, so rewrite it effectively.

Response: The conclusion section was entirely revised (lines 794-804).

Reviewer 3 Report

In this paper authors provides a comprehensive review of the current state of AI-based approaches for lung cancer diagnosis and characterization using histological and cytological images. The introduction effectively highlights the significance of lung cancer, its challenging diagnosis, and the potential role of AI in improving pathologists' routine practice. This approach allows readers to gain insights into how AI can contribute to the accurate classification of different subtypes of lung cancer. Below are my some concern:

·         The linguistic of the paper can be further improved. There are some typos. Some comma and spaces between words are missed. The language used throughout the manuscript needs to be improved

           ·         The motivation of this paper could be clarified. It is important to show the reason for doing this research.

·         Improve the introduction part by making it more concise.

·         The author should supplement the deep learning related literature on 2023 and need to add more works that are relevant.  The Facial Expression Recognition Using Deep Neural Network, Computer vision model with novel cuckoo search based deep learning approach for classification of fish image,  A. Modified Genetic Algorithm with Deep Learning for Fraud Transactions of Ethereum Smart Contract, Image processing model with deep learning approach for fish species classification, The Facial Expression Recognition Using Deep Neural Network, Cuckoo Search-Based Optimization for Cancer Classification: A New Hybrid Approach , A randomized deep neural network for emotion recognition with landmarks detection, A Machine Learning based Approach to Detect the Ethereum Fraud Transactions with Limited Attributes.

            ·         The authors should provide more details on how newcomers in the field, such as research students, can                utilize the current methods for new cases.

            ·         If possible, suggesting possible future research directions can be helpful. The authors can add in section                5, more details and specific suggestions are encouraged the readers. 

            ·         More discussion on the current limitations of deep learning methods can be helpful. Maybe a table that                summarizes them with their main properties - is not a must, just a suggestion.

             ·         The quality of Figures 2 and 3 can be improved, and more graphs should be used to illustrate the results.

Moderate editing of English language required

Author Response

We thank the reviewer 3 for the comments. Underneath, we present the changes made to our manuscript according to the proposed suggestions.

1) The linguistic of the paper can be further improved. There are some typos. Some comma and spaces between words are missed. The language used throughout the manuscript needs to be improved.

Response: The whole manuscript underwent English revisions, and its language is now improved. In addition, we corrected typos and errors related to commas and spaces between words.

2) The motivation of this paper could be clarified. It is important to show the reason for doing this research.

Response: We clearly stated our contribution as well as the main purpose of our study in the last paragraph of the introduction section (lines 85-92). The changes are also highlighted in the text.

3) Improve the introduction part by making it more concise.

Response: We improved our introduction by making it shorter and more concise. In addition, we now clearly state the contribution of our work in the last paragraph of the section. The changes are highlighted in the text.

4) The author should supplement the deep learning related literature on 2023 and need to add more works that are relevant.  The Facial Expression Recognition Using Deep Neural Network, Computer vision model with novel cuckoo search based deep learning approach for classification of fish image,  A. Modified Genetic Algorithm with Deep Learning for Fraud Transactions of Ethereum Smart Contract, Image processing model with deep learning approach for fish species classification, The Facial Expression Recognition Using Deep Neural Network, Cuckoo Search-Based Optimization for Cancer Classification: A New Hybrid Approach , A randomized deep neural network for emotion recognition with landmarks detection, A Machine Learning based Approach to Detect the Ethereum Fraud Transactions with Limited Attributes.

Response:  We updated our systematic review by extending our search in PubMed until 2023 in order to incorporate more recent papers into our revised manuscript (lines 98-100). This procedure identified 25 new articles published in 2023, 9 of which fulfilled our criteria of eligibility and were finally included in our manuscript (References 32-36, 61, 86, 95,96). As a result, the flow diagram (Figure 1) of our paper has been updated (line 142). Information from the newly included articles has been added to the tables and main text of the revised manuscript. Unfortunately, the above-suggested papers could not be retrieved using our search strategy.

5) The authors should provide more details on how newcomers in the field, such as research students, can utilize the current methods for new cases.

Response: In the introduction, we stated that our study aimed to provide a comprehensive guide to advances in the classification, diagnosis, prognosis, and prediction of lung cancer. We have included both histology and cytology to provide a useful summary for both pathologists and cytologists. In the discussion section, which is now revised, more information is provided in Table 8, which emphasizes several limitations of deep learning regarding the proposed methodologies for lung cancer. We also discuss the ethical issues arising from the use of Deep Learning in Pathology (lines 777-804).

6) If possible, suggesting possible future research directions can be helpful. The authors can add in section 5, more details and specific suggestions are encouraged the readers.

Response: We have revised the discussion section by providing additional information about the adoption of Digital Pathology in a pathology laboratory. We also include Table 8, summarizing the main limitations of Deep learning models in cancer classification tasks. Figures 2 and 3 provide an overview of the so far research on lung cancer from a technical point of view. Finally, both in the discussion and conclusion section, we highlight the high-performance metrics of the presented studies, discuss the advantages of deep learning in Pathology and encourage further research in the future. The changes are highlighted in the text.

7) More discussion on the current limitations of deep learning methods can be helpful. Maybe a table that summarizes them with their main properties - is not a must, just a suggestion.

Response: We believe indeed that such a table will provide added value for the discussion of the manuscript, and we thank the reviewer for his suggestion. We added Table 8, which emphasizes several limitations of deep learning regarding the proposed methodologies for lung cancer.

8) The quality of Figures 2 and 3 can be improved, and more graphs should be used to illustrate the results.

Response: The quality of the figures in the revised manuscript has been improved, while Figure 2 has been extended to the whole width of the manuscript to become more distinct.

Round 2

Reviewer 1 Report

I carefully read the revised version of this manuscript. As can be understood, my questions are clarified, and previous issues are resolved. This manuscript is suitable for acceptance.

Reviewer 3 Report

none